# Pocket similarity identifies selective estrogen receptor modulators as microtubule modulators at the taxane site

Yu-Chen Lo[1], Olga Cormier[2], Tianyun Liu[1,3], Kendall W. Nettles[4], John A. Katzenellenbogen[5], Tim Stearns[2,3] & Russ B. Altman[1,3]

Taxanes are a family of natural products with a broad spectrum of anticancer activity. This activity is mediated by interaction with the taxane site of beta-tubulin, leading to microtubule stabilization and cell death. Although widely used in the treatment of breast cancer and other malignancies, existing taxane-based therapies including paclitaxel and the second-generation docetaxel are currently limited by severe adverse effects and dose-limiting toxicity. To discover taxane site modulators, we employ a computational binding site similarity screen of > 14,000 drug-like pockets from PDB, revealing an unexpected similarity between the estrogen receptor and the beta-tubulin taxane binding pocket. Evaluation of nine selective estrogen receptor modulators (SERMs) via cellular and biochemical assays confirms taxane site interaction, microtubule stabilization, and cell proliferation inhibition. Our study demonstrates that SERMs can modulate microtubule assembly and raises the possibility of an estrogen receptor-independent mechanism for inhibiting cell proliferation.

[1] Department of Bioengineering, Stanford University, Stanford, CA, USA. [2] Department of Biology, Stanford University, Stanford, CA, USA. [3] Department of Genetics, Stanford University, Stanford, CA, USA. [4] Department of Integrative Structural and Computational Biology, Scripps Research Institute, Jupiter, FL, USA. [5] Department of Chemistry, University of Illinois-Urbana Champaign, Champaign, IL, USA. These authors contributed equally: Yu-Chen Lo, Olga Cormier. Correspondence and requests for materials should be addressed to R.B.A. (email: Russ.Altman@stanford.edu)

Microtubules are polymers of alpha- and beta-tubulin heterodimers present in all eukaryotic cells[1,2]. Microtubules transport and position cellular components in interphase and form the mitotic spindle in mitosis. Microtubule arrays in both cases are highly dynamic, with the assembly and disassembly of the polymer regulated by the intrinsic tubulin GTP hydrolysis and microtubule-associated proteins[1]. In mitosis, microtubule dynamics ensure the successful capture, alignment, and segregation of chromosomes into the daughter cells. Stabilizing or destabilizing microtubules in mitosis leads to mitotic arrest mediated by activation of the spindle-assembly checkpoint, and, in many cases, apoptotic cell death[3]. As a consequence, targeting the microtubule cytoskeleton has been a successful strategy to treat cancer[4]. One of the most widely used microtubule-stabilizing drugs is paclitaxel (Taxol, Bristol-Myers Squibb), a member of a class of diterpenes identified from the Pacific yew that feature a taxadiene core (taxanes)[5]. Structural studies show that paclitaxel and other taxane compounds interact with the major cleft of beta-tubulin, known as the taxane site, at the inner surface of the microtubule lumen[6]. Binding of paclitaxel to the taxane site induces a conformational change of beta-tubulin that enhances protofilament contacts, leading to microtubule stabilization and suppression of microtubule dynamics[2,6–8]. Although paclitaxel and the second-generation docetaxel (Taxotere, Aventis, Bridgewater, NJ) are two of the most successful chemotherapies for the treatment of breast, ovarian, lung carcinomas, and other malignancies, their clinical use is hampered by drug resistance, hypersensitivity reaction to the drug vehicle, dose-limiting toxicity associated with neurotoxicity, myelosuppression, and other severe side effects[9,10]. Furthermore, most taxane drugs, both semisynthetic analogues of paclitaxel and natural products, have higher molecular weight than paclitaxel, are impractical for oral administration, and offer no improvement in clinical performance over the original compounds[11]. Therefore, identifying a generation of synthetic taxanes remains an attractive strategy for improving the current state of cancer treatment, especially if molecules with optimal pharmacokinetic properties and resistance profiles could be developed rapidly.

A promising strategy for anticancer drug discovery is drug repurposing, also known as drug repositioning, in which a known drug can be repurposed to address cancer indications based on previously off-target interactions[12,13]. Since approved drugs often have optimized transport properties and safety profiles, repurposing known drugs can potentially facilitate drug approval and enable rapid deployment to the clinic. Traditional approaches for drug repurposing are often based on empirical findings from unexpected side effects or through large-scale small molecule screens, which are time-consuming, costly, and do not offer insights into specific drug-binding mechanisms[14]. The recent wide availability of protein crystal structures from the protein data bank (PDB) offers potential opportunities to discover biological activities of known drugs based on detailed structural knowledge of the protein-ligand interaction. Here, we use a structure-based drug repurposing strategy to discover taxane site modulators by evaluating the similarity between the beta-tubulin taxane site and pockets of drug-like compounds. In this study, a computational binding site similarity screen of > 14,000 drug-like pockets from PDB reveal an unexpected similarity between the estrogen receptor (ER) and the beta-tubulin taxane binding pocket. Evaluation of nine selective estrogen receptor modulators (SERMs) via in vivo and in vitro assays confirmed taxane site interaction, microtubule stabilization, and cell proliferation inhibition. Our study demonstrates that SERMs can modulate microtubule assembly as a potential drug repurposing strategy for cancer treatment and suggests a hormone-independent mechanism for inhibiting cell proliferation.

## Results

**Pocket similarity identifies tubulin-ER cross-reactivity.** We performed a structure-based comparison using the PocketFeature algorithm to assess the similarity between the taxane pocket of beta-tubulin and the co-crystal structures retrieved from PDB (Fig. 1a)[15]. To compare two protein binding sites, residues within 6 Å of the co-crystal ligand in the crystal structure were first identified to define a drug-binding pocket[15]. For a given residue in the binding site, the geometric center of the residues was determined based on the location of the heavy atom and a 6 Å microenvironment consisting of 6-concentric radial shells of 480 physical and biochemical descriptors was evaluated around each residue locus[16,17]. The similarity between two microenvironments was determined using a Tanimoto-like score that identified the shared bits between two feature vectors. The pocket microenvironments between two sites were sequentially compared combinatorially to maximize the total matching scores. To quantify the binding site similarity, shared microenvironments between two protein-ligand pockets were used to determine a PocketFeature Score (PFS), which indicates the likelihood that the ligand from the screened protein pockets will interact with the taxane site.

The PFS evaluation between the taxane site and 14,211 ligand binding sites from PDB identified 53 drug-like pockets that shared significant pocket similarity (PFS < −3.5) (Fig. 1b, Method and Supplementary Data 1)[18]. Computational docking of the 53 cognate ligands to the taxane site revealed 36 high-confidence potential binders with predicted binding affinity of less than −6 kcal/mol (Methods and Supplementary Data 2). Target enrichment analysis showed that among the 36 ligands, the most abundant protein target families with sites similar to the taxane pocket were ERs (12), beta-tubulins (4), MAPK14 kinases (4), dihydroorotate dehydrogenases (DHODHs) (3), and 13 other proteins (Fig. 1c). The 12 ER ligands identified from our binding site similarity screen were predominantly SERMs, which are partial agonists of the ERs (Supplementary Data 2)[19,20]. Several SERMs are analogues of raloxifene (RAL) and tamoxifen (TAM), which are FDA-approved drugs for the treatment and prevention of osteoporosis, and for the reduction of breast cancer risk in postmenopausal woman[21]. Other SERMs identified include 7-oxabicyclo[2.2.1]hep-2-ene sulfonate (OBHS) and tetrahydroisoquinoline phenols (THIQP) analogues.

Ligand structure alignment based on protein microenvironments revealed consensus chemical features between paclitaxel and the predicted SERMs. For example, the two hydroxyl phenyl rings of SERMs and paclitaxel were found in a similar position. Our initial structural analysis suggested that these functional groups may contribute to similar intermolecular interactions and cause cross-reactivity between the two targets, unexpected from previous work (Fig. 2a). To further determine if estrogen modulators in general exhibit potential cross-activity to the beta-tubulin, we evaluated the binding site similarity between the taxane site and 144 ERs, co-crystallized with diverse ER ligands, identified from PDB (Supplementary Data 3). To correlate the PFS with chemical structure, the ER ligands were clustered using a network similarity graph in which the nodes are colored based on the computed PFS (red: PFS < −3, blue: PFS > −2) and the edges between nodes represent the chemical similarity between compounds (Tc > 0.75) (Fig. 2b)[22,23]. The chemical similarity network analysis shows that the top SERM ligands with the highest PFS were TAM analogues: 5C7 (1)(−3.56), 5JY (2) (−3.62), 5C6 (3)(−3.88), OBHS analogues: OB3 (4)(−3.54), OB7 (5)(−4.29), RAL analogues: IOG (6)(−3.65), RAL (7)(−3.45), AEJ (8)(−4.13), and THIQP analogues: J0W (9)(−3.76), KE9 (10) (−3.74). The binding pockets of two other known estrogenic compounds, bisphenol-A (BPA) (11)(−2.44) and estradiol (EST)

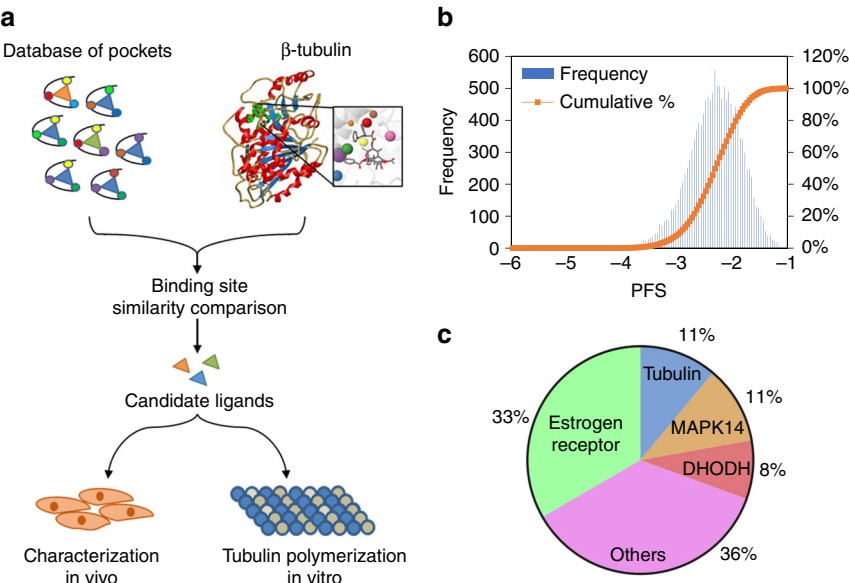

**Fig. 1** In silico binding site similarity screen identifies tubulin-ER cross-reactivity. **a** The computational workflow of identifying and validating ligands binding to the taxane pocket of microtubules. Microenvironments of the taxane pocket were compared to a database of protein pocket microenvironments bound to small molecule ligands. The predicted candidate pockets with the optimal PocketFeature score (PFS < −3.5) were identified and ligands were tested using in vitro tubulin polymerization assay and in vivo cell-based assays. **b** Histogram indicating the distribution of PFS and the cumulative PFS of drug-binding pockets compared to the taxane site from the in silico pocket similarity screen. **c** Target enrichment analysis of the hits identified ER as the most frequent cognate receptor of ligands predicted to bind beta-tubulin

(12)(−2.94) also demonstrated moderate taxane site similarity (Fig. 2c). Overall, our binding site similarity analysis predicts significant compound cross-activity between beta-tubulin and ER and suggests several SERMs have the potential to perturb the microtubule cytoskeleton by interacting with the beta-tubulin taxane site.

**SERMs modulate microtubule organization.** To determine the effects of predicted SERMs on microtubule organization, we evaluated the cell phenotypes treated by selected SERMs 5C7 (1), 5JY (2), 5C6 (3), OB3 (4), and OB7 (5) including three approved drugs RAL (7), LAS (13), and TAM (14), as well as natural estrogen EST (12), on hTERT-RPE1 human epithelial cells, with microtubule stabilizer paclitaxel (TAX)(15) and destabilizer nocodazole (NZO)(16) as controls. The hTERT-RPE1 cells were treated with each SERM at 50 μM and controls, TAX (15) and NZO (16) at 500 nM test concentration for 3 or 18 h; each of the nine SERMs induced microtubule defects to varying degrees, including abnormal mitotic spindles, microtubule bundling, and the formation of cytoplasmic microtubule rings (Fig. 3a, Supplementary Fig. 1). We note that the presence of microtubule rings is an unusual phenotype, and was previously observed with BPA (11), a known estrogen modulator[24]. The observed phenotypes were characterized based on four microtubule morphologies including microtubule bundling, abnormal organization, abnormal spindle formation, and the induction of cytoplasmic rings (Fig. 3b and Supplementary Fig. 2). Microtubule bundling is a characteristic phenotype of microtubule-stabilizing agents (MSAs)[25], presumably due to the action of motors and microtubule-binding proteins on the stabilized microtubules. MSAs also cause abnormal cell organization characterized by changes in cell shape and nuclear position. Dynamic microtubules are an essential component of the mitotic spindle, and abnormal spindle phenotypes such as multipolar spindles, absences of astral microtubules, or mispositioning can also be observed in cells treated with MSAs[26]. To examine the relative penetrance of the microtubule defects resulting from the SERM treatment, cells in each treatment were evaluated for the presence of the four phenotypes and the frequency with which they were observed (average number of abnormalities per cell) (Fig. 3b and Supplementary Data 4). Treatment with RAL (7)(0.91) and LAS (13)(0.87) caused the most severe microtubule defects, followed by TAM derivatives, 5C6 (3)(0.69), 5JY (2)(0.57), and 5C7 (1) (0.52). In comparison, cells treated with EST (12)(0.47), TAM (14)(0.46), and OBHS derivatives, OB7 (5)(0.41) and OB3 (4) (0.33), exhibited only minimal effects on microtubule organization. Importantly, most of these observed phenotypes from the SERM treatment, in aggregate, are similar to those caused by treatment with microtubule-stabilizing agents such as paclitaxel.

**SERMs promote microtubule stability via taxane site binding.** The phenotypes resulting from the SERM treatment suggest that SERMs may enhance microtubule stability. To test this, we evaluated the ability of SERMs to stabilize microtubules using an in vivo nocodazole challenge assay and the compound-induced polymers were visualized using immunofluorescence[27]. hTERT-RPE1 cells were treated with 50 μM of RAL (7), LAS (13), or 500 nM of TAX (15) for 2 h followed by the addition of 16 μM of NZO (16) for 50 min. Three-hour compound treatments resulted in a substantial fraction of cells with microtubules resistant to depolymerization by NZO (16) (Fig. 4a, b), indicating that the tested SERMs were able to confer microtubule stability (Fig. 4a). Consistent with this, cells treated with RAL (7) often have a higher quantity of acetylated microtubules, a modification that is enriched in stable microtubules (Supplementary Fig. 3).

To test whether SERMs are able to affect microtubule polymerization dynamics directly, the nine SERMs were tested for their ability to promote microtubule assembly in vitro. TAX (15) and vinblastine (VLB)(17) were used as positive and negative controls in the fluorescence-based tubulin polymerization assay[28]. Purified tubulin was incubated with test compounds at a final concentration of 50 μM. The degree of tubulin polymerization

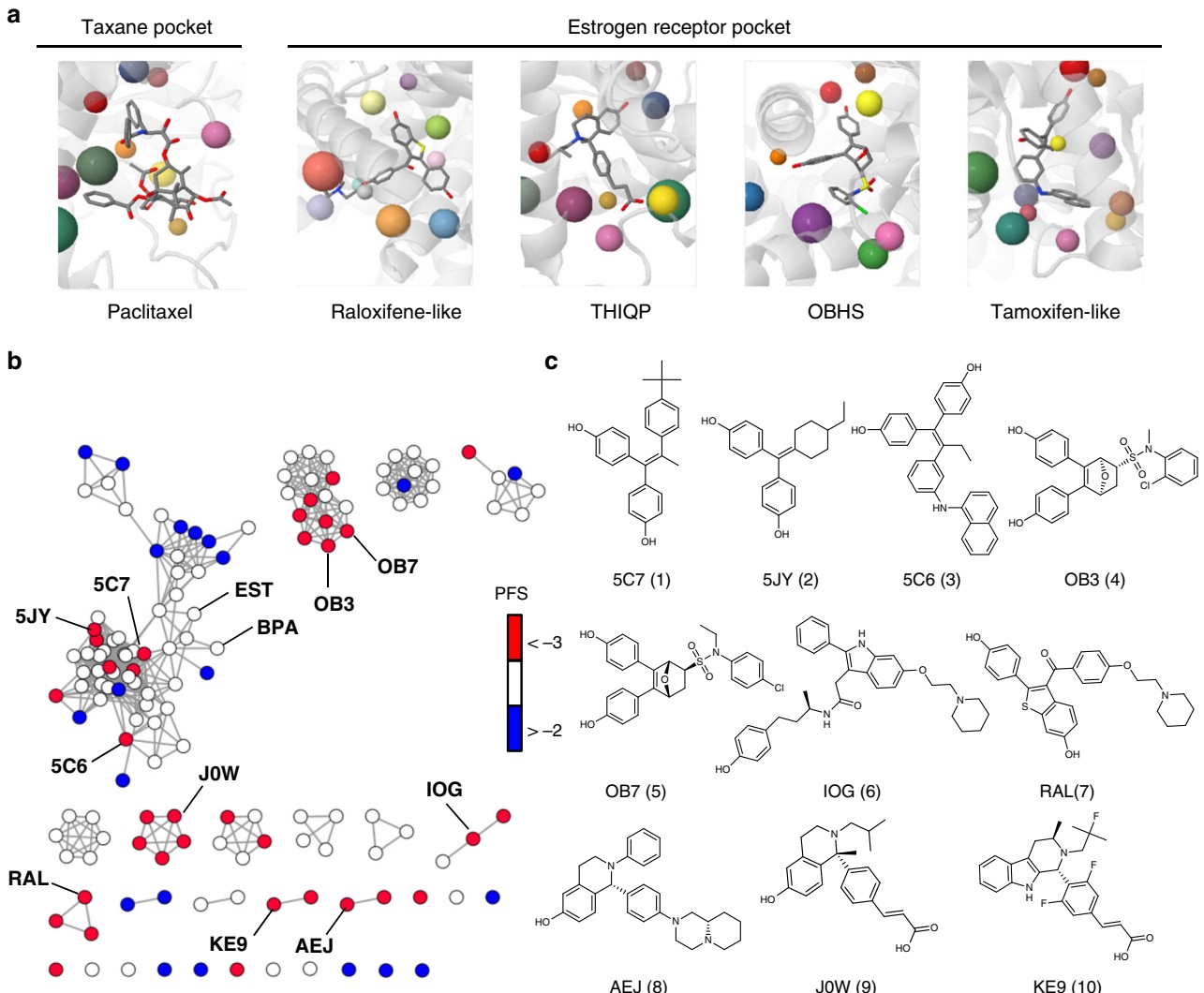

**Fig. 2** Selective estrogen receptor modulators (SERMs) as microtubule modulators. **a** Ligand structure alignment based on protein microenvironments revealed consensus chemical features between TAX (15) and the predicted SERMs within their target pockets (from left to right: PDB: 1JFF [10.2210/pdb1JFF/pdb], PDB: 2R6Y [10.2210/pdb2R6Y/pdb]), PDB: 5FQS [10.2210/pdb5FQS/pdb], PDB: 5KCE [10.2210/pdb5KCE/pdb], PDB: 5DKS [10.2210/pdb5DKS/pdb]). The color spheres represented matching protein microenvironments. **b** Screened estrogen receptor pockets clustered using a network similarity graph: nodes represent ER structures colored based on the computed PFS scores (red: PFS < -3, blue: PFS > -2), edges between nodes represent chemical similarity between the ER ligands (Tc > 0.75). The chemical similarity network analysis shows that the top SERM ligands with the highest PFS were TAM analogues: 5C7 (1), 5JY (2), 5C6 (3), OBHS analogues: OB3 (4), OB7 (5), RAL analogues: IOG (6), RAL (7), AEJ (8), and THIQP analogues: J0W (9), KE9 (10). Two other known estrogenic compounds, bisphenol-A (BPA) (11) and estradiol (EST) (12), whose binding pockets also demonstrated moderate taxane site similarity. **c** Chemical structures of selected SERMs identified from the in silico analysis

was quantified by $V_{max}$ and maximum end-point (MEP) based on the normalized polymerization curves monitored over 30 min and compared their fold changes (FCs) as ratios of SERM over DMSO readout (Fig. 4c and Table 1). Here, an FC threshold value of 1 is used to determine the presence of tubulin polymerization effect relative to DMSO control. Consistent with our in vivo analysis, RAL (7) ($FC_{Vmax} = 2.9$, $FC_{MEP} = 1.7$) showed the strongest enhancement of tubulin polymerization. TAM analogues, 5C7 (1)($FC_{Vmax} = 1.3$, $FC_{MEP} = 1.1$) and 5C6 (3)($FC_{Vmax} = 1.7$, $FC_{MEP} = 1.3$) also enhance tubulin polymerization but to a lesser extent (Fig. 4c). However, 5JY (2)($FC_{Vmax} = 1.1$, $FC_{MEP} = 0.9$), EST (12)($FC_{Vmax} = 0.6$, $FC_{MEP} = 0.6$), and TAM (14)($FC_{Vmax} = 0.4$, $FC_{MEP} = 0.5$) did not promote microtubule formation. Interestingly, OBHS analogues, OB3 (4)($FC_{Vmax} = 1.3$, $FC_{MEP} = 1.1$) and OB7 (5)($FC_{Vmax} = 1.5$, $FC_{MEP} = 1.3$) increased tubulin polymerization in vitro, despite having minimal effect in vivo (Fig. 4c). To ensure that the observed in vitro enhancement of

tubulin polymerization by SERMs was actually due to the formation of microtubules, we visualized assembly products from the polymerization reactions with compounds 5C6 (3), RAL (7), and LAS (13) using negative-stain electron microscopy. The polymers formed were consistent with known microtubule morphology and with those formed by control paclitaxel treatment (Supplementary Fig. 4).

The SERMs tested in this study were selected based on their predicted binding to the taxane site of beta-tubulin, and their ability to enhance polymerization of tubulin and stabilize microtubules is consistent with the action of known taxanes. To test the prediction that the SERMs do bind to the taxane site, we tested the nine selected SERMs for their ability to compete with a fluorescent paclitaxel-conjugate, SiR-Tubulin, for binding to tubulin, using a cell-based imaging assay[29]. To determine the displacement of SiR-Tubulin by the selected SERMs, fluorescence and phase-contrast cell images were compared to identify the

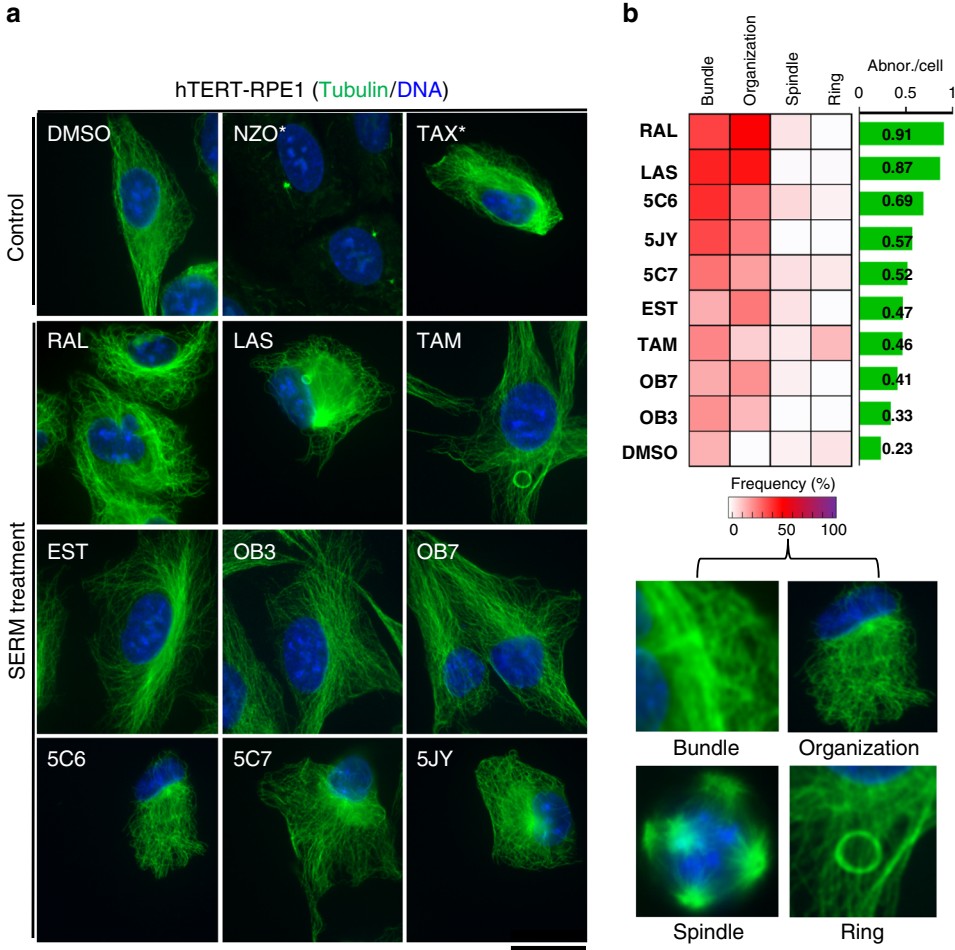

**Fig. 3** SERMs modulate in vivo microtubule organization. **a** hTERT-RPE1 cells were treated with indicated drugs for 3 h at 50 μM (an asterisk (*) indicates treatment with 500 nM) and immunostained for alpha-tubulin (green) and DNA (blue). Scale bar 20 μm. Each of the nine SERMs induced diverse microtubule defects to varying degrees, including abnormal mitotic spindles, microtubule bundling, and the formation of cytoplasmic microtubule rings. **b** The frequency of four specific microtubule effects observed (Supplementary Fig. 2) in each drug condition (from Supplementary Data 4). The average incidence of abnormalities per cells is quantified for each treatment (green bar)

intensity and distribution of the SiR-Tubulin signal (Fig. 4d and Supplementary Fig. 5). Consistent with their microtubule stabilization effect in vitro, the TAM analogues, 5C6 (3) and RAL (7), demonstrated the strongest displacement of SiR-Tubulin fluorescence signal while LAS (13) showed minimal SiR-Tubulin displacement. In contrast, TAM (14), and its derivatives, 5C7 (1) and 5JY (2), OBHS analogues, OB3 (4) and OB7 (5), as well as EST (12), showed no displacement in this assay.

**SERMs inhibit cell proliferation**. Treatment of cells with clinically relevant amounts of paclitaxel inhibits cell growth and ultimately causes cell death[30]. To test whether treatment with the SERMs causes similar effects, we tracked cell proliferation and cell death of hTERT-RPE1 cells during 48 h of SERMs treatment over a range of concentrations, using live-cell imaging (Fig. 5a, Supplementary Fig. 6-7, and Supplementary Movies 1-7). Compound potencies with respect to inhibition of cell proliferation rate ($EC_{50}^{P}$) and cell death were calculated after 24 and 48 h ($EC_{50}^{D24}$ and $EC_{50}^{D48}$). Interestingly, our analysis showed that a broad spectrum of cell death and proliferation dynamics was induced by different classes of SERMs. We found that TAM analogues, 5C7 (1) and 5C6 (3), were strongly cytotoxic, resulting in cell death within several hours of treatment (Fig. 5b). In contrast, RAL (7) and LAS (13) caused a decrease in cell proliferation but little cell

death, and the OBHS analogues, OB3 (4) and OB7 (5) had a negligible effect on proliferation or cell death. To further dissect the mechanism of cell death from the RAL treatment, we examined mitotic cells treated with RAL (7) and DMSO using immunofluorescence microscopy and brightfield time-lapse microscopy (Fig. 5c, Supplementary Fig. 8). Treatment with RAL (7) for 3 h, or longer, resulted in abnormal mitotic spindles in all mitotic cells observed. The mitotic spindles were misoriented both with respect to the normal centered position in the cell and to the normal parallel orientation of the spindle relative to the culture substrate (Fig. 5c). This misalignment is likely due to a reduction in the number of astral microtubules that normally align the spindle.

To test if SERMs can inhibit cell growth of cancer cells, rather than the hTERT-RPE1 cells used in most of our experiments, we analyzed cell proliferation ($GI_{50}$) of selected SERMs from the latest NCI Cancer Screen Data (DTP 60 cell/5 dose-06/2016) (Fig. 5b and Supplementary Data 6)[31]. Since many cancer cell lines express ERs, we further compared $GI_{50}$ values based on the ER status of the lines ($+/-$)(plus: ER positive, minus: ER negative). Except for MCF10a cell line, we note that the ranges of the compound effective concentrations are similar between ER-expressing cells and non-expressing cells. Cells used in this study, hTERT-RPE1 cells, have a similar concentration response based on our assays. Overall, our analysis indicates that the

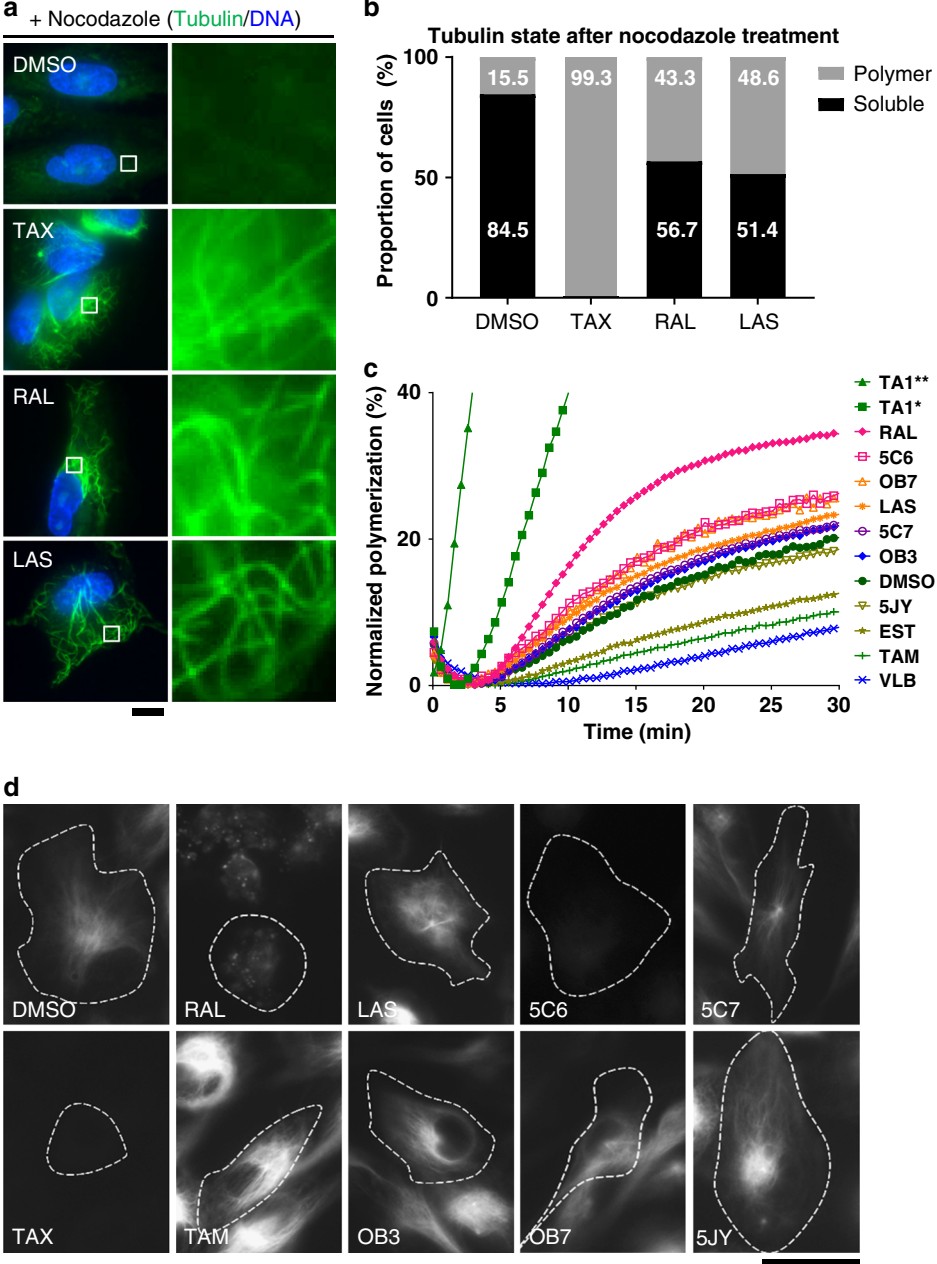

**Fig. 4** SERMs enhance microtubule stability by interacting with the taxane site. **a** hTERT-RPE1 cells were treated with 50 μM of selected SERMs or 500 nM of TAX (15) for 2 h and then subjected to 16 μM of nocodazole challenge for 50 min in the presence of the SERM. The cells were fixed and stained as described. Scale bars 20 μm, inset—800%. **b** Quantification of the proportion of cells in each condition that have microtubules in soluble or polymerized state. **c** Eight selected SERMs: RAL (7), LAS (13), 5C6 (3), 5C7 (1), TAM (14), OB3 (4), OB7 (5), 5JY (2) and controls, TAX (15) and VLB (17), were evaluated for their ability to enhance microtubule polymerization at 50 μM; asterisk (*) indicates treatment with 1 μM. Data were normalized to maximum polymerization readout from 10 μM of paclitaxel treatment (**). **d** hTERT RPE-1 cells were incubated with indicated drugs at 50 μM (except TAX at 2 μM), and SiR-tubulin at 500 nM in the presence of verapamil for 3 h. Representative SiR-tubulin signal for each is shown, outlines indicate cell boundary based on brightfield images (Supplementary Fig. 5). RAL (7), and TAM analogue, 5C6 (3), demonstrated the strongest displacement of SiR-Tubulin fluorescence signal, while LAS (13) showing minimal SiR-Tubulin redistribution. TAM (14), and its derivatives 5C7 (1) and 5JY (2) and OBHS analogues: OB3 (4) and OB7 (5) showed no displacement. Scale bar 35 μm

antiproliferative effect of SERMs on a given cell line does not depend on the ER status of that line.

## Discussion
Approximately 250,000 women in the United States are newly diagnosed with breast cancer each year and SERMs such as RAL (7) and TAM (14) are two of the most commonly used FDA-approved drugs for the prevention of breast cancer or as single or adjuvant chemotherapy[32–34]. Although SERMs are currently recommended only for cancers that are ER-responsive, a recent study showed that TAM treatment was effective in 5-10% of ER-negative breast cancer, suggesting the presence of an alternative cancer-targeting mechanism[35]. In fact, our analysis of the NCI60 data set indicates that the growth of cancer lines that do not express ERs is also inhibited by RAL (7) and other estrogen analogues. Interestingly,

**Table 1 Summary of SERMs effect on microtubules**

| Compound Types | Ligand | Pocket Feature Score | In vitro polymerization[a] | | | | Abn./Cell[b] | In vivo Displacement[c] | Live imaging analysis[d] (µM) | | |
|---|---|---|---|---|---|---|---|---|---|---|---|
| | | | MEP (%) | MEP FC | $V_{max}$ (%/min) | $V_{max}$ FC | | | $EC_{50}^{P}$ | $EC_{50}^{D24}$ | $EC_{50}^{D48}$ |
| | TAX | −10.8 | 100 | 5.0 | 5.8 | 5.9 | 0.96 | Yes | 0.02 | — | — |
| RAL-like | RAL | −3.4 | 34.5 | 1.7 | 2.8 | 2.9 | 0.91 | Yes | 26.8 | 75.9 | 54.2 |
| | LAS | −3.0 | 23.5 | 1.2 | 1.5 | 1.5 | 0.87 | No | 20.8 | 48.1 | 26.7 |
| TAM-like | 5C6 | −3.9 | 26.7 | 1.3 | 1.7 | 1.7 | 0.69 | Yes | — | 23.4 | — |
| | 5C7 | −3.6 | 22.3 | 1.1 | 1.3 | 1.3 | 0.53 | No | — | 26.7 | 27.4 |
| | 5JY | −3.6 | 18.7 | 0.9 | 1.0 | 1.1 | 0.57 | No | 24.4 | 52.8 | 44.6 |
| | TAM | −3.7 | 10.2 | 0.5 | 0.4 | 0.4 | 0.46 | No | 61.3 | 76.4 | 75.1 |
| OBHSN | OB7 | −4.3 | 26.0 | 1.3 | 1.5 | 1.5 | 0.41 | No | 121.4 | — | — |
| | OB3 | −3.5 | 21.9 | 1.1 | 1.3 | 1.3 | 0.33 | No | 133.4 | — | — |
| | EST | −2.9 | 12.7 | 0.6 | 0.6 | 0.6 | 0.47 | No | 39.2 | 78.3 | 77.4 |
| | DMSO | NA | 20.5 | 1.0 | 1.0 | 1.0 | 0.22 | No | NA | NA | NA |

[a]For details, see Fig. 4c, TAX values at 10 µM, other ligands at 50 µM
[b]For details, see Fig. 3a, b, TAX values at 0.5 µM, other ligands at 50 µM
[c]For details, see Fig. 4d
[d]The compound potency on cell proliferation ($EC_{50}^{P}$) and cell death were evaluated after 24 and 48 h ($EC_{50}^{D24}$ and $EC_{50}^{D48}$) based on data seen in Supplementary Figs. 5-6

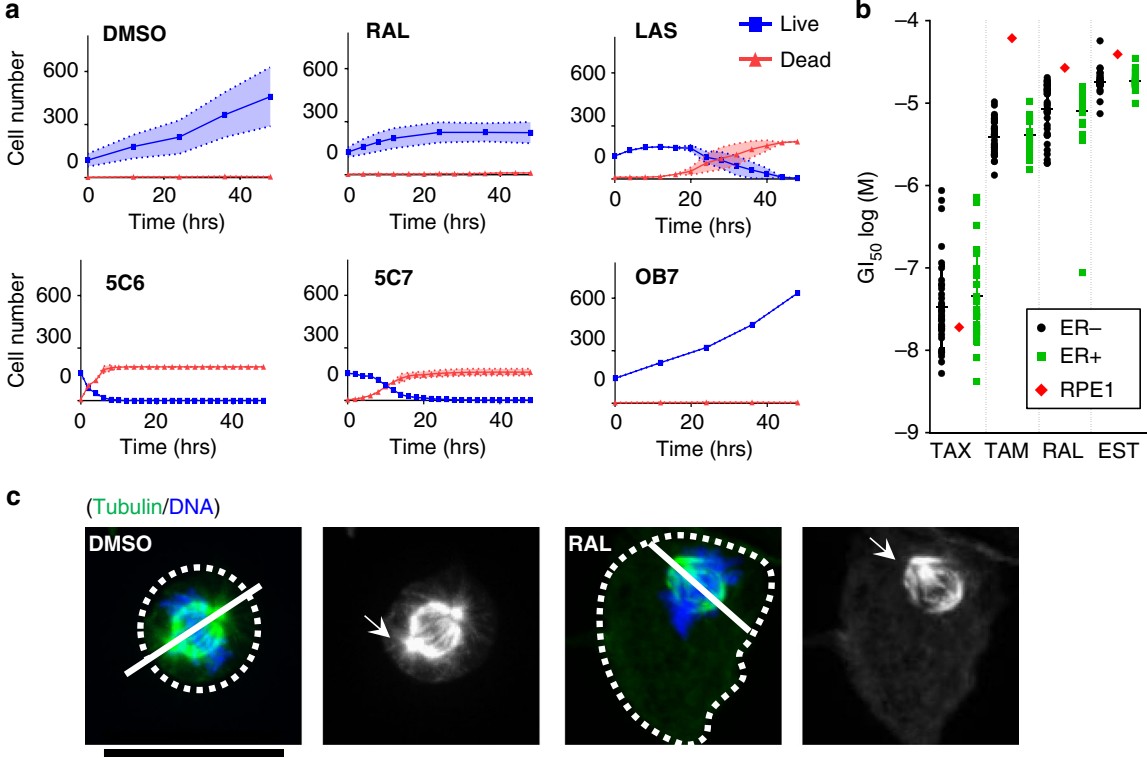

**Fig. 5** SERMs induce mitotic defects and inhibit cell proliferation. **a** The antiproliferative effects of selected SERMs at 50 µM was evaluated by quantifying cells that are alive (blue) and dead (red), as evaluated by the YOYO-3 dye, during 48 h of growth during drug treatment. The mean is shown with error bars representing SEM. **b** Analysis of cancer cells growth inhibition by TAX (15), TAM (14), RAL (7), and EST (12) of NCI60 cell lines with respect to their ER status from the NCI Cancer Screen Data (DTP 60 cell/5 dose-06/2016). Each symbol indicates the mean GI50 of a single cell line for a given drug treatment. Green: cells express ER, black: cells do not express ER, red: $EC_{50}^{P}$ for hTERT-RPE1 cells. **c** hTERT-RPE1 cells were treated with DMSO or RAL (7) for 3 h and mitotic cells were analyzed by anti-tubulin immunostaining (green) and DNA stain (blue). RAL (7) and the mitotic events were tracked using phase-contrast for several hours. Representative mitotic events are shown. Dashed lines indicate the outline of the cells, solid lines indicate the axis of the spindle. Arrows indicate the normal location of astral microtubules. Abnormal spindle organization was observed in all mitotic cells treated with RAL (7) for > 3 h. Scale bar 20 µm

the apoptotic effects of TAM (14) and paclitaxel have been shown to be non-synergistic and non-additive, implicating potential mutual exclusive interactions of these two drug classes with a single protein-ligand binding site[36]. Although several synthetic and natural estrogens including BPA, diethylstilbestrol (DES), 2-methoxyestradiol (2ME), and 17 beta-estradiol (E2-17 beta) have

previously been suggested to perturb microtubule organization, these compounds inhibit microtubule polymerization that is distinct from paclitaxel's mode of action[37–39].

In this work, we used a structure-based repurposing approach to reveal an unexpected pocket similarity between the ER and the beta-tubulin taxane site and identified a mode of action for

SERMs. The selected SERMs fall into several categories based on the in vitro and in vivo properties we have characterized (Table 1). First, both the OBHS analogues, OB3 (4) and OB7 (5), promoted in vitro microtubule polymerization but did not affect in vivo microtubule organization. In contrast, 5JY (2) stabilized microtubules in vivo but did not compete with SiR-Tubulin, suggesting that its observed effects on cell viability and microtubules were not due to direct binding to the taxane site of beta-tubulin, or the tubulin interaction was very transient. On the other hand, our in vitro tubulin polymerization assay showed that EST (12) induced a slight microtubule destabilizing effect. Furthermore, the compound did not induce microtubule bundling phenotype in cell culture nor compete with the SiR-Tubulin binding. This observation is consistent with the known microtubule destabilizing activity of 2-methoxyestradiol, an analog of EST[40]. Importantly, SERMs such as 5C6 (3) and RAL (7) not only demonstrated strong microtubule stabilization effects in the in vitro tubulin polymerization assay but were also capable of displacing of SiR-Tubulin from the taxane site in vivo. As a result, these SERMs, like paclitaxel itself, likely bind directly to the taxane pocket, leading to microtubule stabilization and cell death through a microtubule-mediated mechanism.

The traditional model for cell death caused by an MTA, such as paclitaxel, is microtubule stabilization, followed by cell-cycle arrest and apoptosis[41]. While SERMs like RAL (7) induced microtubule stabilization phenotype during the interphase and defects in mitosis, cell cycle profiling of the compound did not reveal $G_2/M$ cell cycle arrest[13]. Therefore, the cell death by RAL (7) may be due to mitotic slippage similar to low dose paclitaxel treatment[42–44]. On the other hand, 5C7 (1) and 5C6 (3) stabilized microtubules in vitro and produced high incidences of phenotypes typical of disrupted microtubule function. Furthermore, these compounds mediate rapid cell death responses. A possible mechanism for SERM lethality, in this case, is that microtubule stabilization affects microtubule dynamics, which enacts cell death in interphase due to alternative mechanisms related to membrane trafficking, cilium formation, and motility, as suggested in the case of ixabepilone for the treatment of breast cancer[42,45,46]. Furthermore, we believe that the degree of mitotic arrest may be correlated with the ability of SERM and paclitaxel to stabilize microtubules.

High degree of tubulin polymerization from the paclitaxel treatment suggests that paclitaxel has a higher binding affinity to tubulin than SERMs. The difference in their binding affinity can be attributed to paclitaxel's size and structural complexity, which allow the compound to maximally interact with multiple residue contacts within the taxane site[47,48]. On the other hand, SERMs are potential repurposed leads and have not been fully optimized for microtubule binding. Nevertheless, we expect that SERMs may interact with distinct sets of residues in the taxane site that explain SERM-specific phenotypes. Importantly, these features may provide SERMs with potential opportunities to treat cancers that confer resistance from taxane site mutations[49,50]. By determining the structural features critical for SERMs binding, further structure-guided design and medicinal chemistry effort can be directed to improve the affinity and specificity of SERM-tubulin interaction.

While existing microtubule-targeting agents such as paclitaxel are effective treatments against a broad spectrum of cancers, these compounds incur significant adverse events and dose-limiting toxicity in patients. Our study suggests that SERMs such as RAL (7), and certain forms of TAM analogues such as 5C6 (3) should be further explored as taxane site modulators beyond their current indication and could serve as potential drug leads for further development. In addition, our findings strongly implicate a mechanism for ER-independent cell proliferation inhibition during SERM treatment. Understanding what features of SERMs are essential to bind microtubules, affect cell growth or cause

cytotoxicity could lead to better treatment strategies for cancer, and reduction of side effects of current SERM use.

## Methods

**Compound information**. Compounds 1–5 were synthesized as previously described[19]. Compounds 7 and 11–18 were purchased from Sigma Aldrich Inc. All the compounds have been validated by HNMR spectroscopy with > 95% purity. See Supplementary Data 5 for additional compound information.

**Drug-like pocket database creation**. To create the drug-like pocket database, we queried the protein data bank (PDB) (https://www.rcsb.org/) and identified 221485 ligands with at least one co-crystal structure. To identify ligands that satisfied optimal drug-like properties, we filtered the ligands using Lipinski rules, which define a set of criteria for compounds that conform to optimal physicochemical properties, and yielded 56687 ligands with less than 2 Lipinski rules violation and molecular weights from 170 Da to 1000 Da[18]. Further removal of duplicate ligands based on the InChIKey yielded a total of 14217 unique drug-like compounds. Next, corresponding co-crystal receptors of the identified ligands were featurized using the PocketFeature algorithm[15]. Briefly, residues within 6 Å of the ligand were identified as the ligand binding sites. At each residue locus, the PocketFeature program computes the microenvironment of the residue consisting of six concentric shells where each shell is represented by a feature vector of 80 biophysical descriptors. This resulted in a drug-like pocket database where each pocket binding environment is represented by 480-length feature vectors.

**Computational docking of co-crystal ligands**. To validate that the co-crystal ligands of 53 PDB binding pockets identified from the structure similarity search interact with the tubulin taxane site, we computational docked these ligands to the target site of interest using the Autodock Vina program[51]. Briefly, the ligand and receptor input files for the docking program were prepared using the MGLToolKits (http://mgltools.scripps.edu/). To dock the ligand to the receptor site, a bounding box of 40 Å were defined centered on the mean coordinates of the receptor sites. A genetic algorithm was then used to search the optimal ligand binding conformation (pose) within the receptor site points to identify the best pose. Once the docking pose is determined, a docking score (kcal/mol) is evaluated based on the binding interaction energy between the ligand and the receptor. To further estimate the ligand binding affinity (Ki) from the docking score, the following formula was used:

$$\mathrm{Ki} = e^{\left(\frac{\Delta G}{RT}\right)} \qquad (1)$$

where $\Delta G$ is the docking score (kcal mol$^{-1}$), $R = 1.9872036(11) \times 10^{-3}$ kcal K$^{-1}$ mol$^{-1}$, and $T = 300$ K. See Supplementary Data 2 for the docking results.

**Cell culture and immunofluorescence analysis**. hTERT-RPE-1 cells were cultured in DMEM/F-12 (Hyclone, GE Healthcare Life Sciences) supplemented with 10% Cosmic Calf Serum (Hyclone, GE Healthcare Life Sciences). Cells were obtained from ATCC® (CRL-4000), validated morphologically, and periodically tested for mycoplasma and treated if necessary. hTERT-RPE-1 cells were grown on poly-l-lysine–coated #1.5 glass coverslips (Electron Microscopy Sciences). For immunofluorescence analysis, the cells were washed with warmed 80 mM PIPES pH 6.8, 1 mM MgCl2, 5 mM EGTA, 0.5% TX-100 for 30 sec, and fixed with 0.5% glutaraldehyde for 10 min at 37 °C. After fixation, the cells were quenched with 1 mg/mL sodium borohydride in PBS for 10 min at room temperature. Cells were blocked in PBS containing 3% bovine serum albumin (Sigma-Aldrich), 0.1% Triton X-100, and 0.02% sodium azide (PBS-BT). Coverslips were incubated with primary antibodies (mouse anti–α-tubulin (DM1α; Sigma-Aldrich, St. Louis, MO) at 1:4000) diluted in PBS-BT for 1 h at room temperature, washed in PBS-BT, and then incubated in Alexa Fluor dye-conjugated secondary antibodies (Invitrogen) diluted 1:1000 in PBS-BT at room temperature for 30 min. After secondary staining, coverslips were washed in PBS-BT, and nuclei were stained by brief incubation in 4′,6-diamidino-2-phenylindole (DAPI; 1 µg/ml), followed by additional PBS-BT washes. Coverslips were mounted to glass slides using Mowiol (Polysciences) in glycerol containing 1,4,-diazobicycli-[2.2.2]-octane (Sigma-Aldrich) antifade. Wide-field epifluorescence images were acquired using an Axiovert 200 M microscope (Carl Zeiss, Jena, Germany) with PlanApoChromat 63 × /1.4 NA objectives and a cooled, charge-coupled device (CCD) camera (Orca ER; Hamamatsu Photonics, Hamamatsu, Japan). Confocal images of mitotic spindles were acquired using an Axio Observer microscope (Carl Zeiss, Jena, Germany) with a confocal spinning disk head (Yokogawa, Japan), PlanApoChromat ×63/1.4 NA objective, and a Cascade II:512 EMCCD camera (Photometrics). Nocodazole resistance and competition experiment images were acquired using a BZ-X710 acquisition system (Keyence, Osaka, Japan) with a Nikon Plan Apo Brightfield VC ×60/1.40 NA objective. All images were processed using ImageJ (National Institutes of Health, Bethesda, MD) and/or Photoshop (Adobe, San Jose, CA). Microtubule morphologies were scored based on the phenotypes as shown in Supplementary Fig. 2, including those morphologies that appear more than once. The final score is a ratio of the total number of abnormalities observed to the total number of cells observed for each condition.

**Tubulin polymerization assay**. Drug effects on tubulin polymerization were evaluated using tubulin polymerization assay[28]. In brief, ~2 mg/mL tubulin dissolved in tubulin polymerization buffer (80 mM PIPES, 2 mM MgCl2, 0.5 mM EGTA pH 6.9, 30 μM 4', 6-Diamidino-2-phenylindole (DAPI), 1 mM GTP). Cold tubulin preparation was then added to pre-diluted drugs (final concentration 50 μM) or DMSO in a pre-warmed 96-well plate (Corning Costar). Tubulin polymerization was assayed at 37 °C pre-warmed plate reader for 30 min at 30 sec–1 min intervals by measuring the change in fluorescence (Ex = 360 nm; Em = 420 nm). The measurement was done using BioTek SynergyNeo. Data were analyzed using GraphPad Prism version 7.03 (GraphPad Software, La Jolla, CA). In all replicates, 10 μM TAX (**15**) was used as a positive control. To normalize the data and allow comparisons across experiments, the extent of paclitaxel polymerization readout after 30 min is expressed as 100% and the lowest absorbance readout as 0%. Additionally, VLB (**17**) was used as a control for a compound that suppresses tubulin polymer formation[52]. Compounds examined do not cause changes in fluorescence over time when incubated in buffer without tubulin. The degree of polymerization after 30 min was measured as the percent of maximal paclitaxel polymerization readout at 10 μM.

**Electron microscopy**. An in vitro polymerization reaction with 5C6 (3), RAL (7), LAS (13), and TAX (15) was conducted as described above. At the end of 30 min, the reaction was stopped by fixation in 0.5 % glutaraldehyde for 5 min. The microtubules were pelleted and resuspended in polymerization buffer without free tubulin. Resuspended samples were applied to glow-discharged 300 mesh carbon/formvar-coated copper grids (Electron Microscopy Sciences) and allowed to settle for 3 min. The grids were then washed by touching to water drops twice and stained with 1% uranyl acetate for 1 min. The grid was then dried and observed on the JEOL JEM-1400 TEM at 120 kV and images were taken using a Gatan Orius digital camera.

**Nocodazole challenge assay**. Cells were cultured on glass coverslips as described and treated with 50 μM of test compounds, 500 nM TAX (15), or DMSO for 2 h as specified. Subsequently, 16 μM NZO (16) was added to the pre-treated cells for 50 min. Therefore, at the fixation point, cells have been exposed to the test compounds for 2 h 50 min and NZO (16) for 50 min. Cells were fixed and stained as described above. Similar area images were acquired using a BZ-X710 acquisition system (Keyence, Osaka, Japan) with a Nikon Plan Apo Brightfield VC ×60/1.40 NA objective and stitched into composite images using the BZ-accompanying software. Subsequently, images were scored using ImageJ Counter plugin for the presence of microtubule polymer. Cells counted for treatment with DMSO, TAX (15), RAL (7), and LAS (13) were 627, 139, 520, and 138, respectively.

**SiR-tubulin competition assay**. Cells were cultured as described on glass-bottom 96-well dishes (Cellvis). In individual wells, cells were treated simultaneously with 500 nM of SiR-tubulin, a taxol analogue conjugated to silicon-rhodamine (SiR) derivatives (Cytoskeleton Inc.), 10 μM verapamil (Cytoskeleton Inc.) for visualizing microtubules, and tested compounds at a final concentration of 50 μM. As a control, compounds were also tested without the addition of SiR-tubulin to ensure there is no fluorescence. Cells were imaged using a BZ-X710 acquisition system (Keyence, Osaka, Japan) with a Nikon Plan Apo Brightfield VC ×60/1.40 NA objective at 37 °C. To allows sufficient time for SiR-tubulin to penetrate the cells, images were acquired at 3 h post-treatment at the same time point as the immunofluorescence analysis. Following live imaging the cells were fixed with methanol and stained for alpha-tubulin. Same fields of view were acquired, and the cells were identified based on the overall position. In cases where the cells could not be located, a representative cell from the same well is shown.

**Live cell imaging**. Cells were plated in a 96-well Glass (#1.5) Bottom Plate (Cellvis, Mountain view, CA) and allowed to recover for several hours. Drugs and YOYO-3 dye were added at indicated concentrations using a multichannel pipette to reduce the time between pipetting, 20–30 min prior to the image capture of the first time point. Cells were observed using an IncuCyte Zoom dual color live content imaging system (Model 4459, Essen BioSciences, Ann Arbor, USA) residing within a Thermo tissue culture incubator maintained at 37 °C with 5% $CO_2$. Data were acquired using a ×10 objective lens in phase contrast, green fluorescence (ex: 460 ± 20, em: 524 ± 20, acquisition time: 400 ms), and red fluorescence (ex: 585 ± 20, em: 665 ± 40, acquisition time: 800 ms) channels. Data acquisition time per well was negligible. Automated image analysis routines were optimized using the Zoom software package (V2016A/B) to accurately measure confluence over time. In conditions where cell death was within the first 30 min, the confluence measure is omitted due to the excessive cell debris. To determine the number of dead cells, YOYO-3 positive cells were counted using Cell Counter plugin using Fiji distribution of ImageJ 2.0 at each time point. In addition, cells with excessive blebbing were also counted as dead.

**Statistical analysis**. All statistical analysis and graph generation was done using GraphPad Prism version 7.03 for Windows (GraphPad Software, La Jolla, CA). The sample size was chosen to ensure accurate representation of observed phenotypes. Linear regression analysis was used to determine the slope accounting for the rate

of growth for each replicate including standard errors. The slopes were used to interpolate the half growth rate as determined by analyzing the corresponding vehicle controls. Those values are presented at $EC_{50}^P$. For conditions where cell death was observed, $EC_{50}$ was determined for 24 and 48 h, using sigmoidal fitting and used to interpolate the half-death concentrations by analyzing the corresponding controls. Where data are insufficient, $EC_{50}$ was not assigned.

**Reporting summary**. Further information on experimental design is available in the Nature Research Reporting Summary linked to this article.

**Code availability**. The structure-based binding site similarity screen was performed using The PocketFeature program and is available from the Stanford SimTK web server (https://simtk.org/projects/pocketfeature). The chemical similarity analysis was performed using the CSNAP program and is available from the CSNAP web server (http://services.mbi.ucla.edu/CSNAP/).

## Data availability
Data supporting the findings of this manuscript are available from the corresponding author upon reasonable request. A reporting summary for this Article is available as a Supplementary Information file. The drug-like pocket database that supports the findings of this study is available from the Stanford simTK web server (https://simtk.org/projects/serm). The source data underlying Figs. 1b-c, 2b-c, 3, 5b and Supplementary Fig. 2 are provided as a Source Data file.

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

## Acknowledgements

We thank all members of the Helix group and Stearns lab at Stanford University for their helpful feedback and suggestions. We also thank the lab of Scott J. Dixon for live imaging instrument time. The project was supported by a Stanford School of Medicine Dean's Postdoctoral Fellowship, Stanford Graduate Fellowship, and the following funding sources: NIH GM102365, LM05652, HL117798, R01CA220284, and R01GM121424. The project described was supported, in part, by ARRA Award Number 1S10RR026780-01 from the National Center for Research Resources (NCRR). Its contents are solely the responsibility of the authors and do not necessarily represent the official views of the NCRR or the National Institutes of Health.

## Author contributions

Y.-C.L. and R.B.A. initiated the project. Y.-C.L., O.C., T.S. and R.B.A. led the team, designed experiments, and analyzed results. Y.-C.L. performed pocket structure database preparation, binding site similarity screen, target enrichment analysis, and chemical similarity clustering with input from T.L. and R.B.A.. O.C. performed immuno-fluorescence analysis, stabilization analysis, tubulin polymerization assay, competition assay, live-cell imaging, and compound potency evaluation with input from T.S. Compounds were synthesized and prepared by K.W.N. and J.A.K. Manuscript was prepared by Y.-C.L. and O.C. with input from T.S. and R.B.A.

## Additional information

**Competing interests:** The authors declare no competing interests.

