## [Peer Review File · Nature Communications]

Reviewers' Comments:

Reviewer #1:

Remarks to the Author:

The authors used a computational similarity binding-site screen to identify pockets in the PDB that resemble the one on beta-tubulin, which is targeted by microtubule stabilizing agents like the taxanes. They found a similarity between the taxane site and a pocket on the estrogen receptor. Subsequent evaluation of several selective estrogen receptor modulators confirmed their interaction with the taxane site, their microtubule stabilizing activity both in vitro and in cells and their inhibition on cell proliferation. The authors conclude that estrogen receptor modulators could be repurposed for cancer treatment via a hormone-independent mechanism.

This is an interesting and well executed piece of research that reveals a completely unexpected finding of high medical relevance. It thus deserves publication in Nat. Comm. after the authors have addressed the following points:

I am not convinced about the microtubule polymerization data shown Figure 4c. How do the authors know that what they measure is indeed reflecting microtubule formation and not an unspecific protein aggregation? An imaging method like, e.g., negative staining electron microscopy or fluorescence microscopy seems necessary here to clarify this point. Why does taxol seem so much more potent in promoting microtubule formation than the other compounds? Why did the authors not compare similar concentrations of compounds compared to taxol? In this context, I found the exact experimental setup and rationale difficult to follow, including the way the data were normalized. How do the non-normalized data look like?

2-Methoxyestradiol is frequently listed as a colchicine-site microtubule destabilizing agent (see, e.g., reference 5 of the author's manuscript). How does this reconcile with the authors claim that their compound 12 (EST) acts as a microtubule stabilizing agent in vivo?

Regarding the consistency between the Results and Discussion sections a small contradiction was observed. In the Discussion on lines 214 to 216 it is mentioned that EST does not compete with Sir-Tub. However, this data can neither be found in the Results (lines 161 to 170) nor in Figure 4d or Suppl. Figure 3. Furthermore, assuming that only the microtubule polymerization assay is counted as in vitro assay, the results from lines 156 to 158 suggest that TAM, 5JY, and EST do not promote microtubule formation. This contradicts again the statements in the Discussion on lines 214 to 216. The authors should address this issue by either defining more clearly their distinction between the in vivo and in vitro assays or by removing the contradicting wording.

It would be useful for the non-expert reader if the authors could describe in more detail how exactly they performed their computational similarity binding site screen. The corresponding Method section is rather brief.

Reviewer #2:

Remarks to the Author:

This manuscript from Lo et al. utilizes computational modeling and prediction to identify binding pockets of ligands in known receptor-ligand complexes. It then uses the chemical interaction profiles to predict binding pockets and ligands that share similarity with the taxane-binding pocket (and Taxol interaction) on beta-tubulin. This results in a high number of estrogen receptor-ligand complexes among the top hits with shared similarity to tubulin-taxane interactions. The authors then present evidence that a panel of estrogen receptor modulators (SERMs) display activities that are consistent with binding to the beta-tubulin taxane pocket and stabilizing microtubules, and conclude that SERMs act similarly to taxol in stabilizing microtubules and inducing cell death. This suggests SERMs may be lead molecules for the development of microtubule-targeting cancer

therapies and suggests they may inhibit cell proliferation via a hormone-independent mechanism.

Overall this is a very interesting study and utilizes an intriguing approach to identify molecules that may interact productively with the taxane site on beta-tubulin. The approach will be of general interest to a wide audience, and novel ligands for the taxane site hold promise for the development of anti-cancer strategies. The ligand binding analysis is comprehensive and identifies clear hits with tubulins predictably among the top receptors. However, the assessment of SERM activity on microtubules in vivo, and particular in vitro, falls short of being rigorous enough to conclude that SERMs bind directly to the taxane site and stabilize microtubules. In several cases these results present at least the impression of a mostly qualitative assessment. The quantitative analysis of data in these cases should be improved, and more examples provided to allow readers to evaluate the strength of the findings. If SERMs indeed bind to the taxane site and stabilize microtubules, causing cell death, the findings will be highly impactful. Accordingly, the data indicating SERMs promote microtubule polymerization and stabilization in vitro is fundamental to this conclusion and must convincingly demonstrate this activity. If these data can be appropriately strengthened I believe this manuscript would be highly appropriate for publication in Nature Communications.

Major issues:

The authors provide evidence that the SERMs modulate microtubule organization. While this appears to be accurate, the presentation leaves interpretation unclear. What are the scores normalized against? Does taxol = 1 and untreated = 0? Is DMSO considered untreated? The rings are clear, but how are conditions such as 'organization' scored and what are the criteria? It could be helpful to point out examples of various phenotypes and/or include more examples. Putting the concentrations used for the paclitaxel and nocodazole treatments in the main text, near the SERM concentrations, would also help readers evaluate the experiments.

There are instances in which treatment with a SERM was lethal (e.g. LAS treatment overnight in Fig S1). It is unclear if this lethality is being ascribed to the microtubule stabilization, and not clear in the time lapse movies whether the dead cells passed through mitosis. Can the authors clarify the interpretation of SERM lethality?

The authors also present evidence that SERMs interact with the taxane site on beta-tubulin and can stabilize microtubules in vitro. This is critical to support the conclusion that SERMs actually bind with reasonable affinity to the taxane pocket, and do not exert their microtubule-related effects via another mechanism in vivo. This reviewer believes these results, as presented, need to be strengthened:

A) The nocodazole destabilization experiments in Fig 4a are nice. The fact that the RAL and LAS concentrations are presented in the main text and not in the legend, and TAX and Nocodazole concentrations are not presented in the main text, but only in the legend makes the interpretation more convoluted than necessary.

B) The evidence that SERMs promote and/or stabilize microtubules in vitro utilizes only a fluorescence-based assay and does not confirm the state of any polymer formed. Additionally, the quantification of this assay is not fully developed and/or presented. A fluorescence-based assay is a convenient readout, but does not validate whether the SERMs produce microtubules, as opposed to sheets, ribbons or even aggregates. With this assay it is critical to validate that the SERMs produce microtubule polymer, e.g. by negative stain EM. (Another method would be cold depolymerization but this would be hampered by stabilizing drugs in this case. DIC or TIRF based methods can observe individual microtubule dynamics.) The linear increases seen with the 'less active' SERMs is reminiscent of tubulin aggregation at 37C over time. This linear increase is seen with the buffer and DMSO treatments also (and VLB). Additionally, based on the relative increase of the traces in Figure 4c, this steady, linear increase is a large component of the more active

SERMs as well. This baseline rate should be corrected for, but it is unclear if it has been corrected for in the normalized activities reported for each SERM. Also, is the MEP based on the value at 30 min, or based on the theoretical endpoint of the calculated curve (which could be quite different based on the linear increase rather than a plateau at 30 min). It is also unclear why both buffer and DMSO are tested. Are some compounds dissolved in buffer rather than DMSO? DMSO can stimulate microtubule nucleation and should be consistent across all reactions. This is unclear in the methods and presentation. The fact that the taxol trace for 1 μ M is presented in the figure and the data is normalized against 10 μ M taxol does not aid interpretation. Can the trace for 10 μ M be included in the figure? LAS is among the most active compounds in the in vivo phenotype assay in Figure 3, yet very near to DMSO alone in the microtubule assembly assay. Whether SERMs directly promote microtubule stability is critical to the manuscript's conclusions and should be more rigorously demonstrated.

C) The competition assay in Fig 4d is stated to measure the displacement of Sir-tubulin from the taxane site. It is unclear if the Sir-tubulin (0.5 μ M) and SERMs (50 μ M) are added both simultaneously, rapidly one after the other, or if Sir-tubulin is added first and allowed to bind to microtubules before the SERM is added. Only one cell is presented for each case. It appears that DMSO alone displaces Sir-tubulin more effectively than some SERMs (e.g. TAM or OB3). Is this consistently the case? How is this interpreted? Can this assay or a similar competition be quantified?

Taxol is believed to cause death during mitotic arrest or post mitosis due to chromosome segregation errors. The authors note that compounds 5C6 and 5C7 cause cell death within a few hours of treatment (Fig 5a and S5-S6). These compounds produce high incidence of disrupted microtubule phenotypes in fig 3. Yet, this high cytotoxicity appears to occur before cells can undergo mitosis. In the discussion, based largely on the interpretation of Sir-tubulin displacement by 5C6 and not 5C7 in fig 4, the authors conclude that the effects of 5C7 on 'cell viability and microtubules were not due to direct binding to the taxane site'. However, RAL, LAS and 5C6 'bind directly to the taxane pocket and stabilize microtubules, which cause mitotic defects and apoptosis'. The timing of cell death from 5C6 seems inconsistent with the mechanism of paclitaxel. The differences among the various SERMs, not in terms of binding efficiency, but in terms of the range of results obtained with the different assays should be discussed in more detail in the context of what would be predicted for compounds binding to the taxane pocket. Also, if the cytotoxicity of several of the SERMs are concluded not to result from directly binding to the taxane pocket, it becomes more important to demonstrate that the other SERMs do indeed stabilize microtubules in vitro.

The data in Fig 5c is presented to dissect the mechanism of cell death during RAL treatment. These data are difficult to interpret with the current information and presentation. It seems in Fig 5C and main text that RAL inhibits proliferation but caused little cell death. Figure S4 presents two examples of cells treated for relatively short periods (3 to several hours) with RAL and both show blebbing/mitotic failure within minutes of anaphase. Does this cause cytokinesis failure? The generation of a tetraploid progeny would likely inhibit proliferation but may not cause death. How many of the population undergo mitosis in the 48 hours shown in Fig 5a? Yet, dead cells do not accumulate. Perhaps FACS sorting of this population would reveal if they traversed mitosis and failed cytokinesis? The absence of astral microtubules in these spindles is very interesting, and perhaps could also explain cytokinesis problems (e.g. Motegi et al, 2006 Dev Cell 10: 509-20; Paolo et al, 2005 JCS, 118:1549-58). This is the only image of the phenotype and there is no mention of astral microtubules outside the one sentence on page 9. It is also proposed that this may cause the spindle misalignment as astral microtubules normally align the spindle, but no reference is cited. It is also unclear if the alignment is due to the incomplete rounding of the RAL treated cell in Fig 5c, and how the alignment relative to plane of division was scored in these cells. Overall this is an intriguing phenotype but in the current presentation does little to dissect the mechanism of cell death from RAL treatment.

Minor issues:

It may be informative to label RAL and LAS in Figure 2b, as these are two of the most prominent molecules in the study results.

The legend to Figure 1b states that candidates with optimal PFS < -5 were identified and tested. Should this be < -3? If not, please clarify in more detail as it appears there are only 3 hits below -5 PFS.

There are numerous instances of grammatical errors and the text needs an overall careful proofread. For example, the first sentence in the section titled "SERMs modulate microtubule organization" reads: "To determine the effects of SERMS on microtubule organization from our computational prediction, we evaluated the cell phenotype of selected SERMs including three approved drugs RAL, LAS, and TAM on hTERT-RPE1 human epithelial cells, with tubulin stabilizer paclitaxel and destabilizer nocodazole as controls." The phrase 'from our computational prediction' should come after SERMs unless it refers to an idea that the computational prediction also predicted certain effects on microtubule organization (in this case it needs more clarification). The phrase 'evaluated the cell phenotype of selected SERMs' should specify that it is SERM treatment that produces a phenotype. Paclitaxel should be described as a microtubule stabilizer rather than tubulin stabilizer.

Fig S5 legends in the graphs do not have units.

The discussion concludes with the statement "and reduction of side effects of SERM use". The side effects discussed previously result from taxane treatments. Is this a typo or alluding to benefits of optimizing SERM properties to minimize any current side effects that may be due to effects on microtubules?

Reviewer #3:

Remarks to the Author:

Drug repurposing is an interesting approach for the development of new drugs that should receive a greater interest. In this paper the Authors have discovered that some estrogen receptor modulators can also act as taxane site modulators. The compounds have been selected by an in-silico approach and their action was experimentally confirmed.

The paper is well written and in my opinion should be accepted.

Reviewer #4:

Remarks to the Author:

I am commenting solely on the cellular and tubulin studies, which are difficult to interpret since almost all studies were performed with 50 μ M concentrations of the "SERMs," and few attempts were made to determine biologically meaningful concentrations. In my experience with tubulin assays, concentrations above 5-10 μ M are subject to artefactual errors, not least of which is compound precipitation which can distort spectroscopic and fluorescence evaluations. It also seems to me that one of the key requirements for a valid computational methodology is that it produce at least one highly active and unsuspected new compound of interest, and this was not the case for the current manuscript.

Another major defect in the paper is a totally inadequate review of the rather vast literature describing active steroidal compounds that interact with tubulin at both the taxane and colchicine binding sites. The first category includes the rather heroic effort by Susan Mooberry and her group

to define the interactions of the taccalonolides at the taxane site, culminating thus far in taccalonolide AJ; and earlier reports from the Cushman group of 2-ethoxyestradiol analogues that had taxol-like properties. In the second category are the enormous synthetic efforts to improve on the colchicine inhibitory effects of 2-methoxyestradiol that include work by Thurston and colleagues, Potter and colleagues and Cushman and colleagues.

Minor points:

1) SERMs should be spelled out in the title.

2) From Abstract: "via in vivo and in vitro assays." Cells and biochemical assays are both "in vitro" with in vivo being reserved for studies with intact organisms. Better to write "cellular and biochemical assays."

3) The authors use compound abbreviations too haphazardly. It appears that the abbreviation for nocodazole was NOC in the text and apparently NZO in some of the figures. Also, what is LAS? For a number of the compounds under study, the conventional compound 1, compound 2, etc, would be easier for the reader than the lab abbreviations, such as OB3, OB7, etc. Especially as compound numbers are already used in Fig. 2C.

4) The SiR-Tubulin assay needs some explanation. SiR-Tubulin appears to be a fluorescent taxane analogue, rather than a tubulin derivative, but this is unclear from the text. If a tubulin derivative, how does it penetrate the cells? Is SiR-Tubulin commercially available, or did the authors synthesize it?

5) Reaction conditions for the cellular and biochemical experiments were rarely described in detail. What dimethyl sulfoxide concentration was used in each experiment? For Fig. 4c, what was the tubulin concentration? What does "normalization" against the 10 μ M taxol mean? The extent of fluorescence at each time point, or the extent of fluorescence after 30 minutes? What is the buffer curve in Fig. 4c? Representative of assembly or nonspecific fluorescence? Under the reaction conditions used, tubulin should probably not assemble unless contaminated with MAPs or inadequately purified. The vinblastine curve in Fig. 4c is also problematic. Vinblastine at 50 μ M should cause massive spiral formation, unless perhaps the tubulin concentration is very low (< 0.1 mg/mL). The authors need to document that such spirals would not fluoresce, perhaps citing a literature reference. Simpler, however, is using a polymerization inhibitor, such as nocodazole, for this curve.

6) The authors need to demonstrate that their hits, at least RAL and LAS, truly act as taxane site compounds. They must show that the putative polymer has microtubule morphology by electron microscopy; and that, like taxol-induced polymer, it is stable to calcium and to cold. Obviously, for the EM experiments, they would need to use a reaction condition under which microtubules do not form in the absence of taxol or a taxol-like compound.

7) The "nocodazole challenge test" is inadequately described. Is this original with this paper, or is it based on a literature assay. I am familiar with many studies where cells are extracted in a microtubule stabilizing buffer and tubulin quantitated, after extract centrifugation, by PAGE in the supernatant (soluble) and pellet (polymer). Is that what was done here? From the text it appears that microtubules were quantitated by immunofluorescent evaluation of cells. How was this done, and how was partial assembly/disassembly evaluated? I assume there was an initial treatment with taxol, RAL or LAS, followed by nocodazole. How long was each incubation? The authors should note that there are several papers in the literature that show that inhibitors of assembly inhibit taxol-induced assembly. They need to rationalize their observations vs these literature reports.

Reviewers' comments:

Reviewer #1 (Remarks to the Author):

The authors used a computational similarity binding-site screen to identify pockets in the PDB that resemble the one on beta-tubulin, which is targeted by microtubule stabilizing agents like the taxanes. They found a similarity between the taxane site and a pocket on the estrogen receptor. Subsequent evaluation of several selective estrogen receptor modulators confirmed their interaction with the taxane site, their microtubule stabilizing activity both *in vitro* and in cells and their inhibition on cell proliferation. The authors conclude that estrogen receptor modulators could be repurposed for cancer treatment via a hormone-independent mechanism.

This is an interesting and well executed piece of research that reveals a completely unexpected finding of high medical relevance. It thus deserves publication in Nat. Comm. after the authors have addressed the following points:

1) I am not convinced about the microtubule polymerization data shown Figure 4c. How do the authors know that what they measure is indeed reflecting microtubule formation and not an unspecific protein aggregation? An imaging method like, e.g., negative staining electron microscopy or fluorescence microscopy seems necessary here to clarify this point. Why does taxol seem so much more potent in promoting microtubule formation than the other compounds? Why did the authors not compare similar concentrations of compounds compared to taxol? In this context, I found the exact experimental setup and rationale difficult to follow, including the way the data were normalized. How do the non-normalized data look like?

Response: We thank reviewers for the comments. To verify that the microtubule polymerization assay data reflects microtubule formation and not specific protein aggregation, we have performed additional negative staining electron microscopy on three most potent compounds: RAL, LAS and 5C6 and showed that the tubulin formed extended microtubule polymers under compound treatments similar to taxol-treated morphology (see Supplementary Figure 4). The results are consistent with our interpretation that these compounds can enhance microtubule polymerization *in vitro* and did not cause unspecific protein aggregation.

Taxol is much more potent because the compound is a known drug and as a natural product, the size and complexity allow the compound to maximally interact with multiple critical residue contacts in the taxane site. In contrast, SERMs are synthetic small molecules repurposed from off-target interactions. As a potential lead compound, we do not expect SERM to be fully optimized for the taxane site interaction. Nevertheless, we believe that SERMs may interact with the taxane site through distinct sets of residues that may explain new microtubule phenotypes. These features may provide SERMs with potential opportunity to inhibit cancers that confer resistance from

taxane site mutations. By determining the structural features critical for SERM-tubulin interaction, further structure-guided design and medicinal chemistry effort can be directed to improve the binding affinity and specificity of SERM-tubulin interaction. We have added this explanation to the discussion section of the manuscript.

We chose the test concentrations of paclitaxel and SERMs based on the concentrations relevant to MT stabilization observed in cells that appropriately highlights the differences between these compounds. Additional description of the method of normalization has been added to the manuscript (see also tubulin polymerization assay in the method section).

2) 2-Methoxyestradiol is frequently listed as a colchicine-site microtubule destabilizing agent (see, e.g., reference 5 of the author's manuscript). How does this reconcile with the authors claim that their compound 12 (EST) acts as a microtubule stabilizing agent *in vivo*?

Response: We determine the microtubule stabilizing effects of EST *in vivo* based on several microtubule defects including bundles, organization, spindle formation and ring formation. Our observation is that EST does not exhibit strong microtubule stabilization effects including no bundling (one of the major characteristics of microtubule stabilizing agents). Furthermore, the *in vivo* phenotype is consistent with the inability of EST to polymerize MT *in vitro* and their degree of MT polymerization is similar to that of the DMSO control. The difference between EST and 2-Methoxy-EST may be due to the addition of the 2-methoxy functional group which promotes the interaction of the compound to the colchicine binding site (please see reference 22 in the main text).

3) Regarding the consistency between the Results and Discussion sections a small contradiction was observed. In the Discussion on lines 214 to 216 it is mentioned that EST does not compete with SiR-Tub. However, this data can neither be found in the Results (lines 161 to 170) nor in Figure 4d or Suppl. Figure 3. Furthermore, assuming that only the microtubule polymerization assay is counted as in-vitro assay, the results from lines 156 to 158 suggest that TAM, 5JY, and EST do not promote microtubule formation. This contradicts again the statements in the Discussion on lines 214 to 216. The authors should address this issue by either defining more clearly their distinction between the *in vivo* and *in vitro* assays or by removing the contradicting wording.

Response: We thank reviewers for pointing this out. We have now included the SiR-Tub competition assay result for EST in the Supplementary Figure 5 and showed that the compound does not displace SiR-Tubulin. We have also removed the contradicting wording from the discussion.

4) It would be useful for the non-expert reader if the authors could describe in more detail how exactly they performed their computational similarity binding site screen. The corresponding Method section is rather brief.

Response: We thank reviewers for the suggestion and have now provided additional information on the procedure for the computational similarity binding site screen in the result section of the main text. Additional procedures on database preparation and post hit validation using in silico docking can be found in the supplementary information section.

Reviewer #2 (Remarks to the Author):

This manuscript from Lo et al. utilizes computational modeling and prediction to identify binding pockets of ligands in known receptor-ligand complexes. It then uses the chemical interaction profiles to predict binding pockets and ligands that share similarity with the taxane-binding pocket (and Taxol interaction) on beta-tubulin. This results in a high number of estrogen receptor-ligand complexes among the top hits with shared similarity to tubulin-taxane interactions. The authors then present evidence that a panel of estrogen receptor modulators (SERMs) display activities that are consistent with binding to the beta-tubulin taxane pocket and stabilizing microtubules, and conclude that SERMs act similarly to taxol in stabilizing microtubules and inducing cell death. This suggests SERMs may be lead molecules for the development of microtubule-targeting cancer therapies and suggests they may inhibit cell proliferation via a hormone-independent mechanism.

Overall this is a very interesting study and utilizes an intriguing approach to identify molecules that may interact productively with the taxane site on beta-tubulin. The approach will be of general interest to a wide audience, and novel ligands for the taxane site hold promise for the development of anti-cancer strategies. The ligand binding analysis is comprehensive and identifies clear hits with tubulins predictably among the top receptors. However, the assessment of SERM activity on microtubules in vivo, and particular in vitro, falls short of being rigorous enough to conclude that SERMs bind directly to the taxane site and stabilize microtubules. In several cases these results present at least the impression of a mostly qualitative assessment. The quantitative analysis of data in these cases should be improved, and more examples provided to allow readers to evaluate the strength of the findings. If SERMs indeed bind to the taxane site and stabilize microtubules, causing cell death, the findings will be highly impactful. Accordingly, the data indicating SERMs promote microtubule polymerization and stabilization in vitro is fundamental to this conclusion and must convincingly demonstrate this activity. If these data can be appropriately strengthened I believe this manuscript would be highly appropriate for publication in Nature Communications.

Major issues:

1) The authors provide evidence that the SERMs modulate microtubule organization. While this appears to be accurate, the presentation leaves interpretation unclear. What are the scores normalized against? Does taxol = 1 and untreated = 0? Is DMSO considered untreated? The rings are clear, but how are conditions such as 'organization' scored and what are the criteria? It could be helpful to point out examples of various phenotypes and/or include more examples. Putting the concentrations used for the paclitaxel and nocodazole treatments in the main text, near the SERM concentrations, would also help readers evaluate the experiments.

Response: We thank reviewers for the suggestions and have now provided additional details on the calculation of the abnormality scores in the method section. Microtubule morphologies were scored based on the phenotypes as shown in Supplementary Figure 2, including those morphologies that appear more than once. The final score is a ratio of the total number of abnormalities observed to the total number of cells observed for each condition (please see cell culture and immunofluorescence analysis in the method section).

In addition, we have included one example of each phenotype (bundling, organization, spindle, and ring) in figure 3b, Supplementary Figure 2 and explain the criteria for the organization phenotypes in the text. We have included the test concentration of paclitaxel and nocodazole in the main text near the SERM concentration.

2) There are instances in which treatment with a SERM was lethal (e.g. LAS treatment overnight in Fig S1). It is unclear if this lethality is being ascribed to the microtubule stabilization, and not clear in the time lapse movies whether the dead cells passed through mitosis. Can the authors clarify the interpretation of SERM lethality?

Response: We thank reviewers for the feedback and have now included a discussion of SERM lethality in the main text. Our imaging analysis indicated cells that ultimately die may pass through mitosis, have an abnormal mitosis, or not go through mitosis. Since microtubule stabilization can affect microtubule dynamics and microtubule dynamics are critical for events other than mitosis, such as trafficking, cilium formation, and motility, cell death can occur outside of mitotic failure. Furthermore, it has been proposed based on mitotic rates and other data, that taxol has apoptotic effects outside of mitosis based on other microtubule functions and has been discussed by several groups including Komlodi-Pasztor and colleagues (Nat Rev Clin Oncol. 2011. 8(4):244-50.) and Beth Weaver (Mol Biol Cell. 2014. 25(18): 2677–2681).

The authors also present evidence that SERMs interact with the taxane site on beta-tubulin and can stabilize microtubules in vitro. This is critical to support the conclusion that SERMs actually bind with reasonable affinity to the taxane pocket, and do not exert their microtubule-related effects via another mechanism in vivo. This reviewer believes these results, as presented, need to be strengthened:

3) The nocodazole destabilization experiments in Fig 4a are nice. The fact that the RAL and LAS concentrations are presented in the main text and not in the legend, and TAX and Nocodazole concentrations are not presented in the main text, but only in the legend makes the interpretation more convoluted than necessary.

Response: We thank reviewers for pointing this out and have now included the test concentration of suggested compounds in both main text and figure legend.

4) The evidence that SERMs promote and/or stabilize microtubules in vitro utilizes only a fluorescence-based assay and does not confirm the state of any polymer formed. Additionally, the quantification of this assay is not fully developed and/or presented. A fluorescence-based assay is a convenient readout, but does not validate whether the SERMs produce microtubules, as opposed to sheets, ribbons or even aggregates. With this assay it is critical to validate that the SERMs produce microtubule polymer, e.g. by negative stain EM. (Another method would be cold depolymerization but this would be hampered by stabilizing drugs in this case. DIC or TIRF based methods can observe individual microtubule dynamics.) The linear increases seen with the 'less active' SERMs is reminiscent of tubulin aggregation at 37C over time. This linear increase is seen with the buffer and DMSO treatments also (and VLB). Additionally, based on the relative increase of the traces in Figure 4c, this steady, linear increase is a large component of the more active SERMs as well. This baseline rate should be corrected for, but it is unclear if it has been corrected for in the normalized activities reported for each SERM. Also, is the MEP based on the value at 30 min, or based on the theoretical endpoint of the calculated curve (which could be quite different based on the linear increase rather than a plateau at 30 min). It is also unclear why both buffer and DMSO are tested. Are some compounds dissolved in buffer rather than DMSO? DMSO can stimulate microtubule nucleation and should be consistent across all reactions. This is unclear in the methods and presentation. The fact that the taxol trace for 1 uM is presented in the figure and the data is normalized against 10 uM taxol does not aid interpretation. Can the trace for 10 uM be included in the figure? LAS is among the most active compounds in the in vivo phenotype assay in Figure 3, yet very near to DMSO alone in the microtubule assembly assay. Whether SERMs directly promote microtubule stability is critical to the manuscript's conclusions and should be more rigorously demonstrated.

Response: We thank reviewers for the suggestions. To verify that the microtubule polymerization assay data reflects microtubule formation and not specific protein aggregation, we have performed additional electron microscopy on three most potent

compounds: RAL, LAS and 5C6 (see Supplementary Figure 4). The results are consistent with our interpretation that these compounds can enhance microtubule polymerization *in vitro*. We have also clarified the procedure in the methods section. Here, the MEP is based on the value at 30 mins. In addition, we have removed the trace for the buffer. Finally, we have included the trace for 10 μ M taxol in figure 4c.

5) The competition assay in Fig 4d is stated to measure the displacement of Sir-tubulin from the taxane site. It is unclear if the Sir-tubulin (0.5 μ M) and SERMs (50 μ M) are added both simultaneously, rapidly one after the other, or if Sir-tubulin is added first and allowed to bind to microtubules before the SERM is added. Only one cell is presented for each case. It appears that DMSO alone displaces Sir-tubulin more effectively than some SERMs (e.g. TAM or OB3). Is this consistently the case? How is this interpreted? Can this assay or a similar competition be quantified?

Response: We have added the experimental protocol for the competition assay in the materials and methods section. In our competition experiment, the SiR-Tubulin, SERM are added simultaneously and a representative cell is shown for each condition. The compounds such as TAM do not lead to additional staining by SiR-tubulin. Since some bundling can be observed with those compounds (as seen in Figure 3a), the apparent intensity of the SiR-tubulin appears stronger since the microtubules are closer together. However, the DMSO represents the appearance of the signal in cells untreated by compounds and is consistent with completely untreated cells. As such, DMSO does not displace SiR-tubulin. Currently the assay is qualitative, therefore we can only draw conclusions based on the strong effects by compounds such as RAL, and 5C6. Further characterization of displacement dynamics and affinity would be of potential interest.

6) Taxol is believed to cause death during mitotic arrest or post mitosis due to chromosome segregation errors. The authors note that compounds 5C6 and 5C7 cause cell death within a few hours of treatment (Fig 5a and S5-S6). These compounds produce high incidences of disrupted microtubule phenotypes in fig 3. Yet, this high cytotoxicity appears to occur before cells can undergo mitosis. In the discussion, based largely on the interpretation of Sir-tubulin displacement by 5C6 and not 5C7 in fig 4, the authors conclude that the effects of 5C7 on 'cell viability and microtubules were not due to direct binding to the taxane site'. However, RAL, LAS and 5C6 'bind directly to the taxane pocket and stabilize microtubules, which cause mitotic defects and apoptosis'. The timing of cell death from 5C6 seems inconsistent with the mechanism of paclitaxel. The differences among the various SERMs, not in terms of binding efficiency, but in terms of the range of results obtained with the different assays should be discussed in more detail in the context of what would be predicted for compounds binding to the taxane pocket. Also, if the cytotoxicity of several of the SERMs are concluded not to result from directly binding to the taxane pocket, it becomes more important to demonstrate that the other SERMs do indeed stabilize microtubules *in vitro*.

Response: We thank the reviewer for the suggestion. We have now discussed compounds RAL, 5C6, and 5C7 based on the range of assay results we have obtained and their ability to stabilize microtubule to provide an interpretation of SERM lethality in the discussion. The traditional model for cell death caused by an MTA, such as paclitaxel, is microtubule stabilization, followed by cell cycle arrest and cell death. While SERMs like RAL induced microtubule stabilization phenotype during the interphase and defects in mitosis, cell cycle profiling of the compound did not reveal G2/M cell cycle arrest. Therefore, the cell death by RAL may be due to mitotic slippage similar to low dose taxol treatment. On the other hand, 5C6 and 5C7 stabilized microtubules in vitro and produced high incidences of disrupted microtubule phenotypes. Furthermore, these compounds exhibit rapid cell death responses. Our interpretation for SERM lethality in this case is that microtubule stabilization affects microtubule dynamics, which enacts cell death in interphase due to alternative mechanisms related to trafficking, cilium formation, and motility, as suggested in the case of ixabepilone for the treatment of breast cancer.

7) The data in Fig 5c is presented to dissect the mechanism of cell death during RAL treatment. These data are difficult to interpret with the current information and presentation. It seems in Fig 5C and main text that RAL inhibits proliferation but caused little cell death. Figure S4 presents two examples of cells treated for relatively short periods (3 to several hours) with RAL and both show blebbing/mitotic failure within minutes of anaphase. Does this cause cytokinesis failure? The generation of a tetraploid progeny would likely inhibit proliferation but may not cause death. How many of the population undergo mitosis in the 48 hours shown in Fig 5a? Yet, dead cells do not accumulate. Perhaps FACS sorting of this population would reveal if they traversed mitosis and failed cytokinesis? The absence of astral microtubules in these spindles is very interesting, and perhaps could also explain cytokinesis problems (e.g. Motegi et al, 2006 Dev Cell 10:509-20; Paolo et al, 2005 JCS, 118:1549-58). This is the only image of the phenotype and there is no mention of astral microtubules outside the one sentence on page 9. It is also proposed that this may cause the spindle misalignment as astral microtubules normally align the spindle, but no reference is cited. It is also unclear if the alignment is due to the incomplete rounding of the RAL treated cell in Fig 5c, and how the alignment relative to plane of division was scored in these cells. Overall this is an intriguing phenotype but in the current presentation does little to dissect the mechanism of cell death from RAL treatment.

Response: Our imaging analysis reveals that cells treated with SERMs can undergo both morphologically typical or atypical mitoses (as shown in Supplementary Fig 8). Further analysis of cells treated with RAL for 3 hours and imaged for 45 additional hours reveals that within the initial treatment window (3-12hr), cells can experience a prolonged mitosis but resolve it eventually. Cells were not observed to divide after the initial mitosis. In our experiments cells arrested with TAX in a similar fashion, are also able to eventually resolve their arrest but of course in an abnormal fashion. DNA

content evaluation by PI staining and flow cytometry do not indicate any accumulation of cells in the 2n state when treated with RAL. Current data is consistent with a microtubule-mediated mechanism. Detailed mechanistic determination of cell death mechanism by RAL is of potential interest for further investigation.

Minor issues:

8) It may be informative to label RAL and LAS in Figure 2b, as these are two of the most prominent molecules in the study results.

Response: The compounds shown in figure 2b are ligands with a co-crystal structure in PDB. We have replaced ligand LLC with Raloxifene (RAL) in Figure 2b since LLC is a closed analog of RAL but not examined in our study. LAS was not identified from the existing structure complexes and therefore were not labeled in the figure.

9) The legend to Figure 1b states that candidates with optimal PFS < -5 were identified and tested. Should this be < -3? If not, please clarify in more detail as it appears there are only 3 hits below -5 PFS.

Response: We thank reviewers for the suggestion and have now corrected the cutoff score to -3.5 consistent with that in the main text.

10) There are numerous instances of grammatical errors and the text needs an overall careful proofread. For example, the first sentence in the section titled "SERMs modulate microtubule organization" reads: "To determine the effects of SERMS on microtubule organization from our computational prediction, we evaluated the cell phenotype of selected SERMs including three approved drugs RAL, LAS, and TAM on hTERT-RPE1 human epithelial cells, with tubulin stabilizer paclitaxel and destabilizer nocodazole as controls." The phrase 'from our computational prediction' should come after SERMs unless it refers to an idea that the computational prediction also predicted certain effects on microtubule organization (in this case it needs more clarification). The phrase 'evaluated the cell phenotype of selected SERMs' should specify that it is SERM treatment that produces a phenotype. Paclitaxel should be described as a microtubule stabilizer rather than tubulin stabilizer.

Response: We thank reviewers for the feedback and have now corrected the sentences and improved the overall grammar of the text.

Fig S5 legends in the graphs do not have units.

Response: We thank reviewers for the feedback and have now added the units to the legends in the graphs in Fig S5.

11) The discussion concludes with the statement "and reduction of side effects of SERM use". The side effects discussed previously result from taxane treatments. Is this a typo or alluding to benefits of optimizing SERM properties to minimize any current side effects that may be due to effects on microtubules?

Response: Yes, we are alluding to the benefits of optimizing SERM properties to minimize any current side effects that may be due to their binding to microtubules.

Reviewer #3 (Remarks to the Author):

Drug repurposing is an interesting approach for the development of new drugs that should receive a greater interest. In this paper the Authors have discovered that some estrogen receptor modulators can also act as taxane site modulators. The compounds have been selected by an in-silico approach and their action was experimentally confirmed.

The paper is well written and in my opinion should be accepted.

Response: We thank the reviewer for the positive feedback of our manuscript.

Reviewer #4 (Remarks to the Author):

1) I am commenting solely on the cellular and tubulin studies, which are difficult to interpret since almost all studies were performed with 50 μM concentrations of the "SERMs," and few attempts were made to determine biologically meaningful concentrations. In my experience with tubulin assays, concentrations above 5-10 μM are subject to artefactual errors, not least of which is compound precipitation which can distort spectroscopic and fluorescence evaluations. It also seems to me that one of the key requirements for a valid computational methodology is that it produce at least one highly active and unsuspected new compound of interest, and this was not the case for the current manuscript.

Response: We thank reviewers for the feedback. To verify that the microtubule polymerization assay data reflects microtubule formation and not unspecific protein aggregation, we have performed additional electron microscopy on our three most potent compounds: RAL, LAS and 5C6 (please see Supplementary figure 4). The results are consistent with our interpretation that these compounds can enhance microtubule polymerization *in vitro*. We have performed the appropriate controls with running the experiment *in vitro* without tubulin in otherwise same conditions, and do not observe any change in fluorescence without tubulin. There is no data to suggest that compound precipitation present a problem in the current set up.

2) Another major defect in the paper is a totally inadequate review of the rather vast literature describing active steroidal compounds that interact with tubulin at both the taxane and colchicine binding sites. The first category includes the rather heroic effort

by Susan Mooberry and her group to define the interactions of the taccalonolides at the taxane site, culminating thus far in taccalonolide AJ; and earlier reports from the Cushman group of 2-ethoxyestradiol analogues that had taxol-like properties. In the second category are the enormous synthetic efforts to improve on the colchicine inhibitory effects of 2-methoxyestradiol that include work by Thurston and colleagues, Potter and colleagues and Cushman and colleagues.

Response: We thank reviewers for the suggestion and have now added additional literature references on the steroidal compounds that interact with tubulin in the discussion.

Minor points:

3) SERMs should be spelled out in the title.

Response: We have now spelled out SERMs in the title.

4) From Abstract: "via in vivo and in vitro assays." Cells and biochemical assays are both "in vitro" with in vivo being reserved for studies with intact organisms. Better to write "cellular and biochemical assays."

Response: We have now changed the term to "cellular and biochemical assays" in the abstract.

5) The authors use compound abbreviations too haphazardly. It appears that the abbreviation for nocodazole was NOC in the text and apparently NZO in some of the figures. Also, what is LAS? For a number of the compounds under study, the conventional compound 1, compound 2, etc, would be easier for the reader than the lab abbreviations, such as OB3, OB7, etc. Especially as compound numbers are already used in Fig. 2C.

Response: We have now corrected the abbreviation and include both the abbreviation and the compound number in the manuscript. For a mapping between compounds, compound abbreviations, and compound numbers, please see supplementary table 5.

6) The SiR-Tubulin assay needs some explanation. SiR-Tubulin appears to be a fluorescent taxane analogue, rather than a tubulin derivative, but this is unclear from the text. If a tubulin derivative, how does it penetrate the cells? Is SiR-Tubulin commercially available, or did the authors synthesize it?

Response: SiR-Tubulin is a commercially available, previously characterized taxol analog conjugated to silicon-rhodamine (SiR) derivatives and has been previously characterized in the literature (Lukinavičius et al., Nature Methods. 2014. 11:731–733). We have added an explanation for the SiR-Tubulin reagent in the main text.

7) Reaction conditions for the cellular and biochemical experiments were rarely described in detail. What dimethyl sulfoxide concentration was used in each experiment? For Fig. 4c, what was the tubulin concentration? What does “normalization” against the 10 μ M taxol mean? The extent of fluorescence at each time point, or the extent of fluorescence after 30 minutes? What is the buffer curve in Fig. 4c? Representative of assembly or nonspecific fluorescence? Under the reaction conditions used, tubulin should probably not assemble unless contaminated with MAPs or inadequately purified. The vinblastine curve in Fig. 4c is also problematic. Vinblastine at 50 μ M should cause massive spiral formation, unless perhaps the tubulin concentration is very low (< 0.1 mg/mL). The authors need to document that such spirals would not fluoresce, perhaps citing a literature reference. Simpler, however, is using a polymerization inhibitor, such as nocodazole, for this curve.

Response: We thank reviewers for the feedback and we have updated the tubulin polymerization method. The DMSO is used as the same concentration as the tested compound. The tubulin concentration for Fig. 4c is 10 μ M as described in the method. In order to normalize the data and allow comparisons across experiments, the extent of paclitaxel polymerization readout after 30 min is expressed as 100% and the lowest absorbance readout as 0%. We have removed the buffer curve and noted that compounds examined do not cause changes in fluorescence over time when incubated in buffer without tubulin. The appropriate reference has been cited in the text.

8) The authors need to demonstrate that their hits, at least RAL and LAS, truly act as taxane site compounds. They must show that the putative polymer has microtubule morphology by electron microscopy; and that, like taxol-induced polymer, it is stable to calcium and to cold. Obviously, for the EM experiments, they would need to use a reaction condition under which microtubules do not form in the absence of taxol or a taxol-like compound.

Response: To verify that the microtubule polymerization assay data reflects microtubule formation and not unspecific protein aggregation, we have performed additional electron microscopy on three most potent compounds: RAL, LAS and 5C6 (please see Supplementary figure 4). The results are consistent with our interpretation that these compounds can enhance microtubule polymerization *in vitro*.

9) The “nocodazole challenge test” is inadequately described. Is this original with this paper, or is it based on a literature assay. I am familiar with many studies where cells are extracted in a microtubule stabilizing buffer and tubulin quantitated, after extract centrifugation, by PAGE in the supernatant (soluble) and pellet (polymer). Is that what was done here? From the text it appears that microtubules were quantitated by immunofluorescent evaluation of cells. How was this done, and how was partial assembly/disassembly evaluated? I assume there was an initial treatment with taxol, RAL or LAS, followed by nocodazole. How long was each incubation? The authors should note that there are several papers in the literature that show that inhibitors of

assembly inhibit taxol-induced assembly. They need to rationalize their observations vs these literature reports.

Response: We thank reviewers for the feedback and have now added additional details on the nocodazole challenge assay and their quantification in the method section. We used an immunofluorescent-based nocodazole challenge assays where the cells are first incubated with drugs for 2 hours followed by the addition of nocodazole. The nocodazole does not inhibit taxol-induced assembly. We have added additional references for the assay in the main text.

Reviewers' Comments:

Reviewer #1:

Remarks to the Author:

I am happy with how the authors addressed my experimental concerns raised in point 1. However, although presented in the rebuttal, I found it disappointing that the authors did not discuss in the revised manuscript (e.g., in the Discussion section) my concern that 2-methoxyestradiol is generally thought to be a microtubule-destabilizing agent and how this reconciles with the authors claim that their compound 12 (estradiol) acts as a microtubule-stabilizing agent *in vivo*. Concerning point 3, the authors addressed the criticism; however, I was confused to read in line 198 that "LAS showed limited SiR-Tub displacement", which is in contrast to lines 255-257 where it is stated that "5C6, RAL and LAS were also capable of displacing 257 of SiR-Tubulin from the taxane site *in vivo*."

Reviewer #2:

Remarks to the Author:

The revised manuscript from Lo et al. has been improved in several ways. However, there remain a few substantial issues that were not adequately addressed, and should be addressed prior to publication in Nature Communications.

Major issues:

1) The most serious issue is the quantification of the *in vitro* microtubule polymerization reported in Fig 4c. The conclusions of this study are based on the idea that SERMs bind to the taxane site and stabilize microtubules. It is not accurate to normalize the polymerization assays to the lowest absorbance value to determine the maximum end-point (MEP). This gives the impression that compounds like, for example, LAS (13) and 5C7 (3) have MEP values of 23.5 and 22.3% of 10 μ M Taxol activity, respectively. In the discussion LAS is reported to have 'strong microtubule stabilizing effects in the *in vitro* tubulin polymerization assay' (line 256), and 5C6 is interpreted to have 'stabilized microtubules *in vitro*' (line 266). In reality, 5C6 and LAS induce polymerization only ~2% and ~4% more than DMSO alone. Since the sample compounds contain the same amount of DMSO, all sample reactions should be normalized against the activity of DMSO alone. Thus, the amount of microtubule polymerization and/or stabilization by these compounds is much less than the apparent amount when normalized against no polymerization (lowest absorbance value). The DMSO control is also not included on Table 1, but is an important control as all reactions contain the same amount of DMSO. One may argue that vinblastine is the correct value for no polymerization, but this simply means that values should be normalized against this curve, and DMSO alone remains very close to activity seen with several SERMs. This correction will significantly alter the reported activity for these compounds and require an appropriate adjustment in the discussion.

2) Acknowledging that the SiR-tubulin competition assay is qualitative, presenting essentially a single cell for each condition does not facilitate interpretation. For instance, in both Fig. 4d and Fig. S5 the LAS treatment appears to resemble treatment with EST, OB7, 5JY and 5C7. Yet it is reported to be more effective. Also, the SiR-tubulin and alpha-tubulin panels may not be aligned properly in Fig. S5, as the cells do not seem to correspond in all panels (e.g. TAM, LAS).

Minor issues:

11) The discussion concludes with the statement "and reduction of side effects of SERM use". The side effects discussed previously result from taxane treatments. Is this a typo or alluding to benefits of optimizing SERM properties to minimize any current side effects that may be due to effects on microtubules?

Response: Yes, we are alluding to the benefits of optimizing SERM properties to minimize any current side effects that may be due to their binding to microtubules.

Perhaps a slight modification would be helpful, for instance, to insert the word 'current' to clarify to readers: "and reduction of side effects of current SERM use".

Reviewers' comments:

Reviewer #1 (Remarks to the Author):

I am happy with how the authors addressed my experimental concerns raised in point 1. However, although presented in the rebuttal, I found it disappointing that the authors did not discuss in the revised manuscript (e.g., in the Discussion section) my concern that 2-methoxyestradiol is generally thought to be a microtubule-destabilizing agent and how this reconciles with the authors claim that their compound 12 (estradiol) acts as a microtubule-stabilizing agent *in vivo*. Concerning point 3, the authors addressed the criticism; however, I was confused to read in line 198 that "LAS showed limited SiR-Tub displacement", which is in contrast to lines 255-257 where it is stated that "5C6, RAL and LAS were also capable of displacing 257 of SiR-Tubulin from the taxane site *in vivo*."

Response: We thank reviewers for the feedback. The estradiol (EST) (compound 12) is not a microtubule stabilizing agent based on our *in-vitro* or *in-vivo* assay result. In fact, our *in-vitro* tubulin polymerization assay data showed that EST induced a slight microtubule-destabilizing effect (Figure 4c). Furthermore, the compound did not cause microtubule bundling in cell culture nor compete with SiR-Tubulin binding (Figure 3a and Supplementary Figure 5). The observation is consistent with the microtubule-destabilizing effect of 2-methoxyestradiol, an analog of EST. To clarify this, we have added this explanation to the discussion. In addition, we have now modified our sentence in 257 (now line 260) to indicate that LAS did not displace SiR-Tubulin from the taxane site *in vivo*.

Reviewer #2 (Remarks to the Author):

The revised manuscript from Lo et al. has been improved in several ways. However, there remain a few substantial issues that were not adequately addressed, and should be addressed prior to publication in Nature Communications.

Major issues:

1) The most serious issue is the quantification of the *in vitro* microtubule polymerization reported in Fig 4c. The conclusions of this study are based on the idea that SERMs bind to the taxane site and stabilize microtubules. It is not accurate to normalize the polymerization assays to the lowest absorbance value to determine the maximum endpoint (MEP). This gives the impression that compounds like, for example, LAS (13) and 5C7 (3) have MEP values of 23.5 and 22.3% of 10 μ M Taxol activity, respectively. In the discussion LAS is reported to have 'strong microtubule stabilizing effects in the *in vitro* tubulin polymerization assay' (line 256), and 5C6 is interpreted to have 'stabilized microtubules *in vitro*' (line 266). In reality, 5C6 and LAS induce polymerization only

~2% and ~4% more than DMSO alone. Since the sample compounds contain the same amount of DMSO, all sample reactions should be normalized against the activity of DMSO alone. Thus, the amount of microtubule polymerization and/or stabilization by these compounds is much less than the apparent amount when normalized against no polymerization (lowest absorbance value). The DMSO control is also not included on Table 1, but is an important control as all reactions contain the same amount of DMSO. One may argue that vinblastine is the correct value for no polymerization, but this simply means that values should be normalized against this curve, and DMSO alone remains very close to activity seen with several SERMs. This correction will significantly alter the reported activity for these compounds and require an appropriate adjustment in the discussion.

Response: We thank the reviewer for the suggestion. To help compare the relative microtubule stabilizing effect of our compounds relative to the DMSO control, we have now replaced the MEP and Vmax values with their respective fold change (FC) in the result section by computing the ratios of MEP/Vmax over DMSO readout. The original MEP and Vmax values are reported in Table 1. Using a FC threshold value of 1, we showed that RAL ($FC_{MEP}=1.7$, $FC_{Vmax}=2.9$), 5C6 ($FC_{MEP}=1.3$, $FC_{Vmax}=1.7$), OB3 ($FC_{MEP}=1.1$, $FC_{Vmax}=1.3$), OB7 ($FC_{MEP}=1.3$, $FC_{Vmax}=1.5$), and OB3 ($FC_{MEP}=1.1$, $FC_{Vmax}=1.3$) have observed microtubule stabilizing effect while 5JY ($FC_{MEP}=0.9$, $FC_{Vmax}=1.1$), EST ($FC_{MEP}=0.6$, $FC_{Vmax}=0.6$), and TAM ($FC_{MEP}=0.5$, $FC_{Vmax}=0.4$) has minimal to no effect. This correction is consistent with our previous observation on the relative degree of microtubule polymerization of SERMs.

2) Acknowledging that the SiR-tubulin competition assay is qualitative, presenting essentially a single cell for each condition does not facilitate interpretation. For instance, in both Fig. 4d and Fig. S5 the LAS treatment appears to resemble treatment with EST, OB7, 5JY and 5C7. Yet it is reported to be more effective. Also, the SiR-tubulin and alpha-tubulin panels may not be aligned properly in Fig. S5, as the cells do not seem to correspond in all panels (e.g. TAM, LAS).

Response: We agree with the reviewer and have now modified our sentence indicating that LAS has minimal competition with Sir-Tubulin in cell culture. The differences between the panels in Fig. S5 may be due to the use of methanol fixation, which causes slight cell morphology changes. We have indicated this in the text and figure legends of Fig. S5 in the manuscript.

3) Minor issues:

1) The discussion concludes with the statement "and reduction of side effects of SERM use". The side effects discussed previously result from taxane treatments. Is this a typo or alluding to benefits of optimizing SERM properties to minimize any current side effects that may be due to effects on microtubules?

Response: Yes, we are alluding to the benefits of optimizing SERM properties to minimize any current side effects that may be due to their binding to microtubules.

Perhaps a slight modification would be helpful, for instance, to insert the word 'current' to clarify to readers: "and reduction of side effects of current SERM use".

Response: We have now inserted the word "current" to the sentence to make this clarification to the reader.

Reviewers' Comments:

Reviewer #1:

Remarks to the Author:

The authors have addressed all my concerns and I am happy to support publication.

Reviewer #2:

Remarks to the Author:

The revised manuscript from Lo et al. has sufficiently addressed all concerns except for Major Issue #1. In this case, although the use of fold change (FC) is described in the text, it would be prudent for readers evaluating these data to include a DMSO row in Table 1. For instance, under Ligand heading could be 'DMSO Control' with Pocket Feature Score = NA, MEP (%) = ~ 20.5, MEP FC = 1.0, Vmax (%/min) = 1.0, Vmax FC = 1.0. This would allow the reader to place data in context while viewing the table without going and referencing the text.

It would also be very helpful to indicate in footnote #1 of Table 1 that TAX is used at either 10 uM or 1 uM, and each tested ligand is at 50 uM. Importantly, the current table does not specify whether the 1 or 10 uM Taxol curve is used in the table. For that matter, it would be quite informative to include values from both TAX polymerization assays (1 & 10 uM) in the table.

REVIEWERS' COMMENTS:

Reviewer #1 (Remarks to the Author):

The authors have addressed all my concerns and I am happy to support publication.

Reviewer #2 (Remarks to the Author):

The revised manuscript from Lo et al. has sufficiently addressed all concerns except for Major Issue #1. In this case, although the use of fold change (FC) is described in the text, it would be prudent for readers evaluating these data to include a DMSO row in Table 1. For instance, under Ligand heading could be 'DMSO Control' with Pocket Feature Score = NA, MEP (%) = ~ 20.5, MEP FC = 1.0, Vmax (%/min) = 1.0, Vmax FC = 1.0. This would allow the reader to place data in context while viewing the table without going and referencing the text.

It would also be very helpful to indicate in footnote #1 of Table 1 that TAX is used at either 10 uM or 1 uM, and each tested ligand is at 50 uM. Importantly, the current table does not specify whether the 1 or 10 uM Taxol curve is used in the table. For that matter, it would be quite informative to include values from both TAX polymerization assays (1 & 10 uM) in the table.

Response: We thank reviewer #2 for the comment. We have now added one DMSO row to the table 1 as suggested. In addition, we have indicated the taxol test concentration in the footnote for each assay.